# A polygenic score method boosted by non-additive models

Rikifumi Ohta [1,4] ✉, Yosuke Tanigawa [2,3,4] ✉, Yuta Suzuki [1], Manolis Kellis [2,3] ✉ & Shinichi Morishita [1] ✉

Dominance heritability in complex traits has received increasing recognition. However, most polygenic score (PGS) approaches do not incorporate non-additive effects. Here, we present GenoBoost, a flexible PGS modeling framework capable of considering both additive and non-additive effects, specifically focusing on genetic dominance. Building on statistical boosting theory, we derive provably optimal GenoBoost scores and provide its efficient implementation for analyzing large-scale cohorts. We benchmark it against seven commonly used PGS methods and demonstrate its competitive predictive performance. GenoBoost is ranked the best for four traits and second-best for three traits among twelve tested disease outcomes in UK Biobank. We reveal that GenoBoost improves prediction for autoimmune diseases by incorporating non-additive effects localized in the MHC locus and, more broadly, works best in less polygenic traits. We further demonstrate that GenoBoost can infer the mode of genetic inheritance without requiring prior knowledge. For example, GenoBoost finds non-zero genetic dominance effects for 602 of 900 selected genetic variants, resulting in 2.5% improvements in predicting psoriasis cases. Lastly, we show that GenoBoost can prioritize genetic loci with genetic dominance not previously reported in the GWAS catalog. Our results highlight the increased accuracy and biological insights from incorporating non-additive effects in PGS models.

Predicting heritable traits and genetic liability of disease from individuals' genomes has important implications for tailoring medical prevention and intervention strategies in precision medicine. Polygenic score (PGS), a statistical approach, has recently attracted substantial attention due to its potential relevance in clinical practice[1–3]. To estimate genetic predisposition, PGS aggregates the effects of multiple genetic variants into a single score for each individual. Most existing PGS approaches benefit from the increased sample size in modern genome-wide association studies (GWAS). Specifically, summary-statistics-based approaches take GWAS-based univariate effect size estimates of additive effects and ancestry-matched linkage-

disequilibrium (LD) reference panels as input to construct multivariate predictive models, as implemented, for example, in clumping and thresholding (C + T, also known as pruning and thresholding)[4], LDpred[5], lassosum[6], PRS-CS[7], and SBayesR[8].

Genetic dominance is well-documented in the classical genetics literature. It refers to any deviation from additive effects of allelic dosage of a genetic variant on a trait and consists of dominant, recessive, over-dominant, and over-recessive effects. Each non-additive effect differs by the relationship among the estimated effect sizes for heterozygous or homozygous genotypes (Supplementary Methods). Earlier studies focused on a smaller number of samples and

[1]Department of Computational Biology and Medical Sciences, Graduate School of Frontier Sciences, The University of Tokyo, Kashiwa, Chiba, Japan. [2]Computer Science and Artificial Intelligence Laboratory, Massachusetts Institute of Technology, Cambridge, MA, USA. [3]Broad Institute of MIT and Harvard, Cambridge, MA, USA. [4]These authors contributed equally: Rikifumi Ohta, Yosuke Tanigawa. ✉e-mail: ricky.ohta@edu.k.u-tokyo.ac.jp; tanigawa@mit.edu; manoli@mit.edu; moris@edu.k.u-tokyo.ac.jp

estimated that the relative contribution of non-additive effects on trait heritability is smaller than that of additive effects[9]. However, the increase in sample sizes in recent studies coupled with methodological advancements started to demonstrate the presence of non-additive heritability, and their relative contribution varies depending on the genetic architecture of traits[10–14]. For example, some highlight substantial roles of non-additive effects in autoimmune diseases, most notably in rheumatoid arthritis and psoriasis[15], suggesting the potential benefits of considering non-additive effects in polygenic score modeling for some traits. Indeed, several case studies have already started to explore PGS, focusing on non-additive effects. For example, a recent study reported that replacing additive GWAS *p*-values with recessive GWAS *p*-values in C + T PGS modeling improves predictive performance for aggressive behavior[16]. However, the advantage of incorporating non-additive genetic dominance effects in PGS modeling across a wide range of traits is largely unknown, partly because most existing PGS approaches are limited and consider only linear additive effects, given their dependencies on the GWAS summary statistics characterized by additive association tests.

Here, we overcome the technical limitations and present Geno-Boost, the first computationally efficient and flexible polygenic modeling framework capable of considering both additive and non-additive genetic dominance effects in a unified framework. To account for genetic dominance not captured in additive GWAS summary statistics, GenoBoost directly operates on individual-level data, building on top of the recent methodological innovations by us and others on PGS modeling, including penalized regression[17–25] and statistical boosting[26–30], both relying on individual-level data. Among those emerging approaches, we hypothesize that a statistical boosting framework would be ideally suited for integrating additive and non-additive genetic dominance effects in PGS, given its modularity and flexibility, robustness in high-dimensional problems, and its high predictive ability across a wide range of machine learning problems spanning several decades[26,31–33]. We develop an iterative algorithm for GenoBoost, where we find the genetic effects of homozygous and heterozygous minor alleles relative to homozygous major alleles using analytical solutions (Supplementary Methods), enabling computationally efficient inference on large-scale genetic datasets consisting of hundreds of thousands of individuals. We demonstrate the competitive predictive performance of GenoBoost by benchmarking our approach against seven widely used PGS approaches. GenoBoost shows competitive predictive performance across twelve disease outcomes in UK Biobank, including the best predictive performance of four traits and second best for three phenotypes. We also highlight that the genetic loci prioritized by GenoBoost are often supported by experiments in the literature, suggesting the utility of our approach to capturing biologically validated effects. Overall, our work highlights the advantage of GenoBoost in incorporating non-additive genetic dominance effects in polygenic prediction.

## Results

### Overview of the GenoBoost study
With GenoBoost, we fit a PGS model using individual-level data in an iterative procedure. We first adjust for the covariate effects and gradually expand the set of genetic variants included in the model in the subsequent iterative steps (Fig. 1). We designed the GenoBoost algorithm using a statistical boosting framework[26], where we construct a PGS predictor as a sum of weak predictors, each of which considers only one genetic variant as an input and returns a score based on the individual's genotype for the variant. In each iteration, we focus on the residualized phenotypes adjusted for the effects of covariates and genetic variants already selected in the previous iteration steps and select the most informative genetic variant in predicting the residualized phenotype (Fig. 1a, step 1). To calculate the genotype-dependent scores in the weak predictor for the selected variant for homozygous

major and minor alleles and heterozygous alleles (denoted as $s_0$, $s_2$, and $s_1$, respectively), we take advantage of our analytical solution on the provably optimal genotype-dependent scores (Supplementary Methods), thus enabling the application of GenoBoost to large-scale datasets consisting of hundreds of thousands of individuals and more than one million genetic variants (Methods). We derive the analytical solutions in Theorem 1-7 in Supplementary Methods. Following the theoretical foundations and the best practices in constructing boosting models on high-dimensional datasets while minimizing the risk of overfitting[34], we shrink the optimal scores by the learning rate, $\gamma$ ($0 < \gamma \leq 1$), and iteratively update the GenoBoost model with the regularized scores (Fig. 1a, step 2). The number of iterations and the learning rate are the hyperparameters of the GenoBoost model, and we optimized them using five-fold cross-validation, where we randomly split the model development set into five folds to define training and validation sets (Fig. 1a, step 3, Fig. 1b, Methods). We used the held-out test set for the predictive performance evaluation. We considered two types of GenoBoost models: Additive GenoBoost and Non-additive GenoBoost. The former considers additive effects alone, whereas the latter considers additive and non-additive genetic dominance effects. We used the validation set to select the best-performing model between the two and reported it as the GenoBoost model (Methods).

We assembled a panel of twelve disease outcomes with known heritable basis and high prevalence in UK Biobank and systematically applied GenoBoost to each trait using 1,073,318 imputed biallelic single nucleotide variants (SNVs) for $n = 338,138$ unrelated white British individuals (Methods, Fig. 1b, Supplementary Fig. 1)[35,36]. Using the held-out test set individuals, we evaluated the predictive performance of the resulting model using four metrics: (1) covariate-adjusted pseudo-$R^2$ as in the literature[7,37,38], (2) odds ratios of disease incidence rates stratified by the top 1, 3, 5, and 10% of PGS value, which measure the ability of PGS models in stratifying the individuals with high genetic liability of the disease, (3) area under the receiver operating characteristic curve (AUC), and (4) area under precision-recall curve (AUPRC) (Methods, Fig. 1c). We considered all five models from the five-fold cross-validation when evaluating the predictive performance metrics. Subsequently, we randomly selected one of them to interpret the genetic variants selected in the model (Methods). We inferred the mode of genetic inheritance (i.e., additive, dominant, recessive, overdominant, or over-recessive) of the selected genetic variants based on the relationship between the three weights for the variants assigned by GenoBoost (Methods, Fig. 1d, Supplementary Fig. 2). We compared our classification of the genetic effects against the ones from an orthogonal approach based on additive and non-additive GWAS analyses. We sought literature support for the biologically relevant roles of genes annotated as the closest genes for the selected variants with genetic dominance effects (Methods, Fig. 1e).

### Benchmarking GenoBoost across twelve disease outcomes in UK Biobank
We applied GenoBoost and seven previously published PGS methods to the twelve disease outcomes and evaluated their predictive performance (Table 1). In the panel of twelve disease outcomes in UK Biobank[35,36], we included seven commonly studied traits (rheumatoid arthritis, inflammatory bowel disease, asthma, atrial fibrillation, breast cancer, coronary artery disease, and type 2 diabetes) in the PGS literature[5,7,8,20,39–41] and five disease outcomes (psoriasis[42], gout[43], all-cause dementia[44], Alzheimer's disease[45], and colorectal cancer[46]) with high prevalence and genetic basis (Methods)[36]. Among the twelve traits selected, rheumatoid arthritis and psoriasis are included in the list of candidate phenotypes with substantial non-additive genetic dominance effects in a recent study[15]. Our selection of twelve disease outcomes has a minimum overlap of case individuals among others, except for the moderate overlap between Alzheimer's disease and all-cause dementia (Jaccard index = 0.41) (Supplementary Fig. 3). We fit

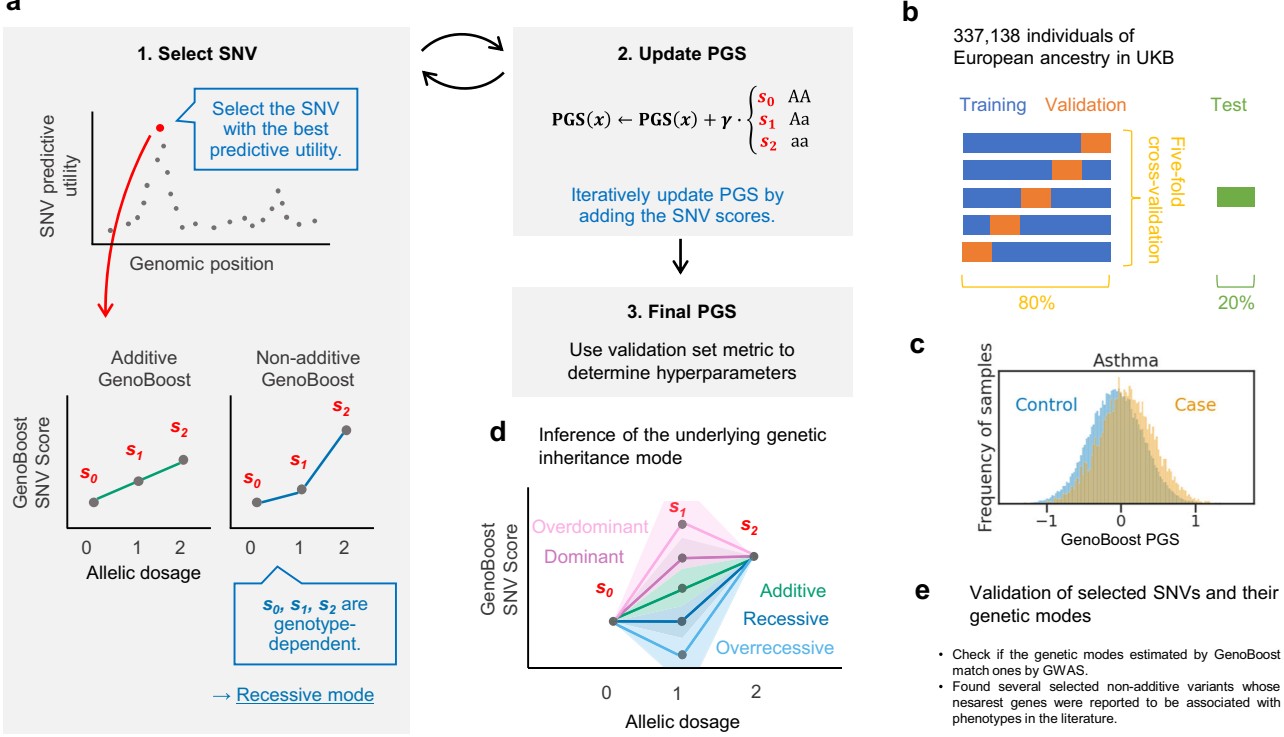

**Fig. 1 | Schematic overview of the study. a** GenoBoost algorithm fits a polygenic score (PGS) function in an iterative procedure. In each iteration, GenoBoost selects the most informative SNV for trait prediction conditioned on the previously characterized effects and characterizes the genotype-dependent GenoBoost scores, $s_0$, $s_1$, and $s_2$. We considered two types of GenoBoost models: Non-additive GenoBoost and Additive GenoBoost, where an additional constraint among the three Geno-Boost scores ensures non-additive genetic dominance effects are always set to zero. GenoBoost iteratively updates its model using two hyperparameters: learning rate $\gamma$ ($0 < \gamma \le 1$) and the number of iterations. We optimized the hyperparameters based on the predictive performance in the validation set using five-fold cross-validation.

**b** Model development and evaluation dataset. We randomly split the unrelated white British individuals in UK Biobank into training, validation, and test sets. We applied five-fold cross-validation. **c** Predictive performance evaluation of PGS models. We used covariate-adjusted pseudo-$R^2$ as the primary metric of predictive performance. **d** We inferred the mode of inheritance of each genetic variant based on the deviation from the linearity in the three GenoBoost scores (Methods). **e** We showed that the inferred genetic inheritance is consistent with GWAS-based approaches. We applied GenoBoost to prioritize genetic loci with genetic dominance not previously reported in the GWAS catalog.

each model five times using the five training sets from the five-fold cross-validation, and we evaluated the median of the predictive performance metrics in the held-out test set individuals (Fig. 2, Supplementary Fig. 4, Supplementary Data 1).

**Table 1 | The panel of twelve disease outcomes in UK Biobank analyzed in the study**

| Phenotype | $n_{case}$ | $n_{control}$ | case prevalence |
|---|---|---|---|
| Rheumatoid arthritis | 8040 | 329,098 | 2.38% |
| Psoriasis | 6595 | 330,543 | 1.96% |
| Gout | 9462 | 327,676 | 2.81% |
| Inflammatory bowel disease | 3703 | 333,435 | 1.10% |
| Asthma | 48,234 | 288,904 | 14.31% |
| All-cause dementia | 4987 | 332,151 | 1.48% |
| Alzheimer's disease | 2060 | 335,078 | 0.61% |
| Atrial fibrillation | 25,839 | 311,299 | 7.66% |
| Breast cancer | 14,870 | 181,027 | 7.59% |
| Colorectal cancer | 6869 | 330,269 | 2.04% |
| Coronary artery disease | 23,184 | 313,954 | 6.88% |
| Type 2 diabetes | 25,589 | 311,549 | 7.59% |

For each phenotype, the number of case and control individuals and case prevalence in the unrelated white British individuals are shown.

Overall, GenoBoost demonstrated a competitive predictive performance, showing the best predictive performance for four phenotypes and the second-best predictive performance for three phenotypes. No single method ranked the best across all tested traits when we evaluated covariate-adjusted pseudo-$R^2$, suggesting the best-performing PGS methods depend on the genetic architecture of the trait (Fig. 2a). GenoBoost and LDpred[5] ranked the best predictive performance across four out of twelve phenotypes each (rheumatoid arthritis, psoriasis, gout, and inflammatory bowel disease for Geno-Boost and atrial fibrillation, breast cancer, colorectal cancer, and coronary artery disease for LDpred), positioning them as the best-performing models. For example, GenoBoost showed the best predictive performance for inflammatory bowel disease (covariate-adjusted pseudo-$R^2$ = 0.00796) with 2.2% improvements over lassosum[6], the second-best PGS method (pseudo-$R^2$ = 0.00778); LDpred showed the best predictive performance for breast cancer (pseudo-$R^2$ = 0.0286) followed by snpboost[27] (pseudo-$R^2$ = 0.0277) and GenoBoost (pseudo-$R^2$ = 0.0263). For the remaining four phenotypes, snpnet[20] was the best PGS method for all-cause dementia and Alzheimer's disease, and snpboost and PRS-CS[7] were the best for asthma and type 2 diabetes, respectively.

We found that the PGS models from GenoBoost have the least number of genetic variants (Fig. 2d, Supplementary Fig. 4e, Supplementary Data 1) while maintaining competitive predictive performance. The sparsity of the GenoBoost model is advantageous in interpreting the genetic loci contributing to the prediction, which we

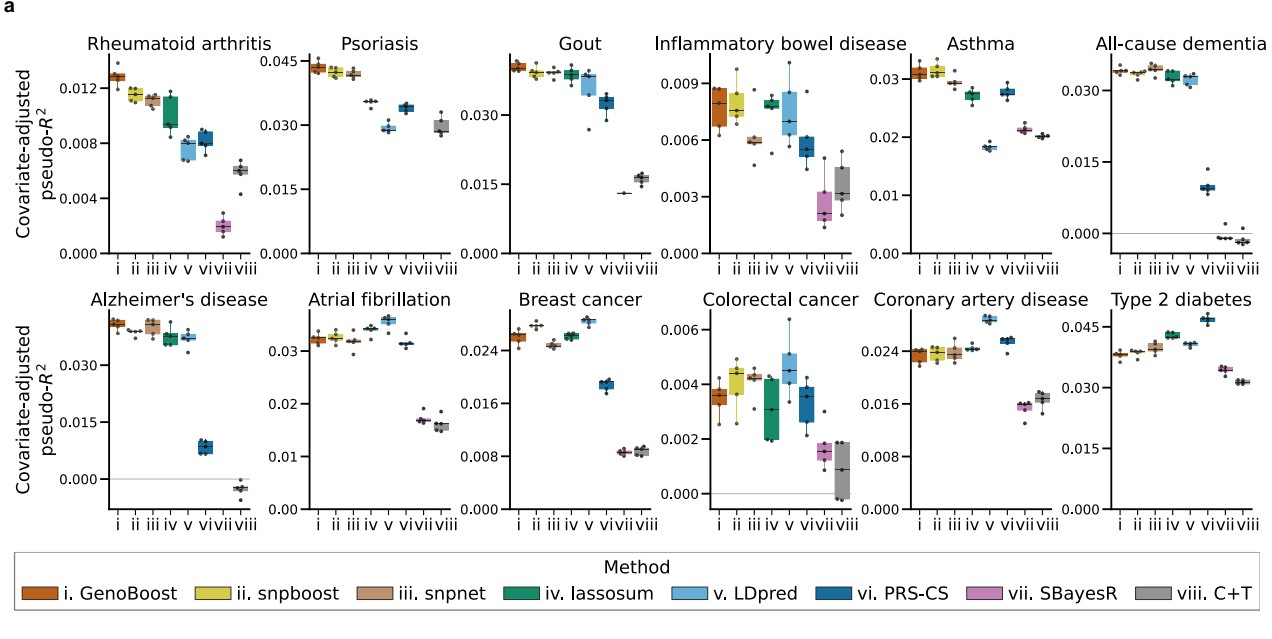

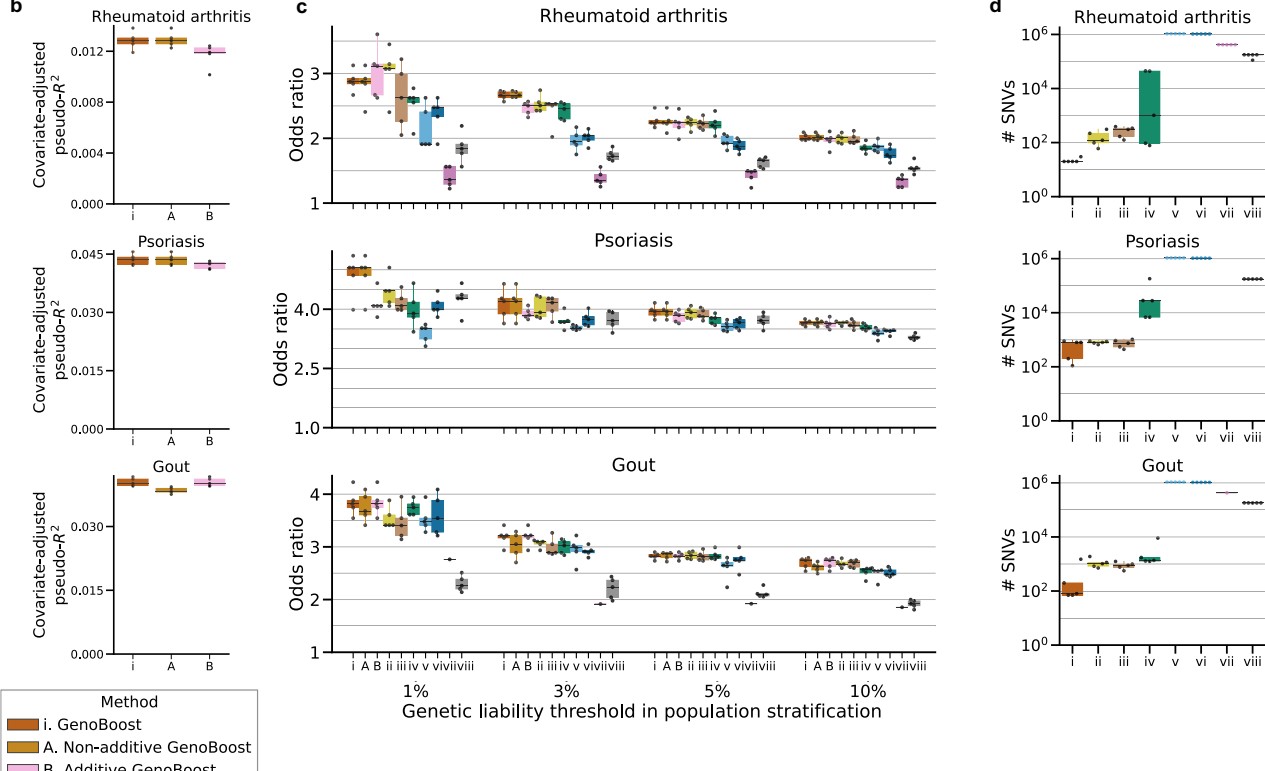

**Fig. 2 | Benchmarking GenoBoost across twelve disease outcomes in UK Biobank. a** Predictive performance in covariate-adjusted pseudo-$R^2$ across eight PGS methods (i-viii, including GenoBoost) and twelve disease outcomes in UK Biobank. C + T: clumping and thresholding. **b** Comparison of Additive (method A) and Non-additive (method B) GenoBoost in their predictive performance across three immune-related disorders: rheumatoid arthritis, psoriasis, and gout. The GenoBoost PGS (method i) was selected between the two based on the predictive performance in the validation set. **c** Comparison of the eight genotype-only PGS models in their ability to stratify case individuals in the top 1, 3, 5, or 10 percentile (x-axis) of the predicted PGS scores in the held-out test set. We show the odds ratio comparing the stratified individuals and the remainder of the population.

**d** Number of selected genetic variants (y-axis) in eight PGS models. The y-axis is shown in a log scale. The results from five-fold cross-validation ($n = 5$ folds) are shown in the plots. The box plots show the median and interquartile range (IQR, defined as the difference between the third and the first quartile points, i.e., Q3 and Q1) with whiskers (Q1-1.5 * IQR and Q3 + 1.5 * IQR). SBayesR raised convergence error for psoriasis and Alzheimer's disease and for four out of five cross-validations of gout. Source data are provided as a Source Data file.

will use to examine the factors influencing the best PGS models as we investigate in the next section.

## The impacts of non-additive effects at the MHC locus and polygenicity on the best PGS models

We focused on three immune-related disorders, where GenoBoost showed the best predictive performance with the most improvements over the second-best methods. Specifically, our Non-additive GenoBoost performed the best for the two autoimmune diseases, rheumatoid arthritis and psoriasis, with covariate-adjusted pseudo-$R^2$ of 0.0129 and 0.0436, showing 11.3% and 2.78% improvements over snpboost[27], the second-best performing PGS methods (Fig. 2b, Supplementary Fig. 4a, Supplementary Data 1). Additive GenoBoost was ranked the best for gout (pseudo-$R^2 = 0.0401$), showing 2.09% improvements over snpboost.

We found that the improvements in the predictive performance for autoimmune diseases with GenoBoost can be attributed to non-additive genetic dominance effects in a relatively small number of genetic loci. For example, genetic variants in the major histocompatibility complex (MHC) locus in chromosome 6 contribute substantially to predicting the genetic liability of autoimmune diseases in the GenoBoost model. The heterozygosity of the MHC regions is protective against psoriatic arthritis[47], and some reports indicate over-dominant selection at the locus[48]. To evaluate the contribution of non-additive genetic dominance effects at the MHC locus, we prepared additional sets of Additive and Non-additive GenoBoost models for autoimmune disease outcomes without using genetic variants on chromosome 6. Specifically, we removed genetic variants on chromosome 6 from GenoBoost models and additionally constructed another set of GenoBoost models without considering chromosome 6 in model training. In both cases, we found that the predictive performance of Additive and Non-additive GenoBoost models are largely consistent (Supplementary Fig. 5, Supplementary Data 1, Supplementary Note 1), highlighting the unique ability of GenoBoost to incorporate non-additive genetic dominance effects at the MHC locus in improving genetic risk prediction of autoimmune diseases.

We hypothesized that the moderate regularization of non-additive genetic dominance effects in GenoBoost would further improve the predictive performance, given the lower frequency of homozygous carriers in populations and limited roles of non-additive effects beyond the MHC locus. We introduced the maximum absolute value for the GenoBoost $s_2$ score, representing the effect of homozygous minor alleles in the GenoBoost model. We indeed found that the modest regularization improved the predictive performance (Supplementary Figs. 6, 7, Supplementary Data 1).

We also found that the polygenicity of the trait also influences the best-performing PGS models, and our sparse GenoBoost PGS model performs the best for less polygenic traits. We first investigated the results for psoriasis and asthma, where GenoBoost ranked second (pseudo-$R^2 = 0.0308$) after snpboost[27] (pseudo-$R^2 = 0.0310$), given the known role of the MHC locus in both traits and the difference between the two traits in the polygenicity (Supplementary Fig. 8). The distribution of the PGS scores in the held-out test set individuals also reflects the difference in polygenicity (Supplementary Fig. 9). To further test the effects of polygenicity on the predictive performance, we generated 80 synthetic phenotypes under eight simulation configurations with varying levels of polygenicity and heritability, applied GenoBoost and LDpred[5], the best-performing method without the sparsity constraints, and compared their predictive performance (Supplementary Fig. 10, Supplementary Data 1, Supplementary Note 2). GenoBoost outperformed LDpred for seven out of eight parameter configurations tested in our simulation analysis. The difference in the predictive performance between GenoBoost and LDpred is larger in less polygenic scenarios. In our benchmarking on the twelve UK Biobank disease outcomes, LDpred was ranked as the

best-performing PGS method for highly polygenic traits, such as atrial fibrillation, breast cancer, and coronary artery disease, suggesting that sparse PGS methods like ours are most advantageous for less polygenic traits.

## Comparison among PGS methods on the individual-level data

We next compared GenoBoost against snpnet[20] and snpboost[27], the two recently developed PGS methods that directly operate on individual-level data. The comparison demonstrated the unique advantage of GenoBoost in incorporating non-additive genetic dominance effects and also resulting in extremely sparse PGS models. Across twelve traits, the three PGS methods on the individual-level data have fewer genetic variants than those based on summary statistics (Fig. 2d, Supplementary Figs. 4e, 8, Supplementary Data 1). Indeed, GenoBoost selected the least number of genetic variants for ten out of twelve traits. Among the three methods on the individual-level data, GenoBoost has the highest level of sparsity with the median of 195 genetic variants, while snpnet and GenoBoost selected the median of 913 and 727 genetic variants across twelve traits.

We compared GenoBoost against snpboost[27], given that both methods are built on statistical boosting. Overall, GenoBoost showed improved predictive performance over snpboost across eight out of twelve traits, in which our methodological advancements in statistical boosting likely played a substantial role. Specifically, our theoretical results on the analytical solution for optimal GenoBoost score enabled computationally efficient PGS modeling while allowing us to consider both additive and non-additive genetic dominance effects.

We showed 4.9% average improvements across the twelve traits over snpnet[20], an implementation of batch screening iterative lasso on large-scale genetic datasets while considering additive effects alone (Fig. 2a). For Alzheimer's disease and all-cause dementia, snpnet showed the best predictive performance (covariate-adjusted pseudo-$R^2 = 0.0408$ and 0.0344, respectively), followed by GenoBoost with a small difference in predictive performance (pseudo-$R^2 = 0.0408$ and 0.0342, respectively), highlighting the competitive performance of GenoBoost. To quantify the benefits of non-additive genetic dominance effects in a penalized regression framework, we considered a variant of snpnet capable of considering both additive and non-additive effects and evaluated the predictive performance (Supplementary Methods). We found that non-additive snpnet models improved prediction for rheumatoid arthritis and psoriasis over additive snpnet models (Supplementary Fig. 11, Supplementary Data 1, Supplementary Note 3). Nonetheless, GenoBoost outperformed non-additive snpnet, highlighting its competitive advantage.

## Stratifying individuals with high genetic liability of diseases

We next demonstrated the ability of GenoBoost to stratify individuals with the high genetic liability of immune-related disorders within a population. Specifically, we used GenoBoost to predict genetic liability of the disease outcomes in the individuals in the held-out test set, evaluated the enrichment of disease prevalence in the identified high-risk group over that in the rest, and summarized the enrichment of case prevalence as odds ratio (Fig. 2c, Methods). For psoriasis, for example, where GenoBoost showed the covariate-adjusted pseudo-$R^2$ of 0.0436, individuals in the top 1% genetic liability predicted by GenoBoost have five-fold enrichment in case prevalence compared to the remainder of the population (odds ratio=5.05). When compared against the snpboost[27] and snpnet[20], the second-best and the third-best predictors with covariate-adjusted pseudo-$R^2$ of 0.0424 and 0.0417 and odds ratio of 4.46 and 4.08, GenoBoost showed 2% and 13% improvements in stratifying psoriasis cases based on predicted genetic liability. We observed similar competitive advantages of GenoBoost across other immune-related disorders, such as rheumatoid arthritis and gout, and at different threshold levels, including the top 3%, 5%, and 10%. When applied across all twelve traits, we found that the top-

ranked PGS by odds ratio is largely consistent with that by covariate-adjusted pseudo-$R^2$ (Supplementary Fig. 4d, Supplementary Data 1). We also confirmed that other commonly used metrics of binary classification accuracy, including AUC and AUPRC, capture consistent patterns (Supplementary Fig. 4a-c, Supplementary Data 1). We observed a limited predictive performance of GenoBoost in non-European population groups in UK Biobank, as in other PGS methods trained only on genetic datasets from European individuals (Supplementary Fig. 12, Supplementary Data 1, Supplementary Note 4). We observed substantial variability in predictive performance in non-European populations across disease outcomes and PGS methods. For example, GenoBoost showed the best predictive performance for psoriasis for the East Asian population and gout for the South Asian population, whereas other methods showed better performance for gout for the African population and type 2 diabetes for the South Asian population, motivating further expansion of GenoBoost into multi-ancestry settings in future studies (Discussion). Together, those results highlight the competitive predictive performance of GenoBoost.

### GenoBoost scores allow inference of the mode of inheritance

The three GenoBoost scores for each genetic variant ($s_0$, $s_1$, and $s_2$) in the sparse GenoBoost PGS models allow us to classify the genetic variant effects based on the types of the estimated inheritance mode into additive, dominant, recessive, over-dominant, and over-recessive (Supplementary Fig. 2, Supplementary Methods). Applying this technique, we inferred that 40–67% of genetic variants selected by GenoBoost show non-zero non-additive genetic dominance effects, of which 10-22% of all of the selected genetic variants have non-zero recessive effects (Fig. 3a). For example, our approach classified 53% (16 of 30 variants) and 67% (602 of 900 variants) of the selected genetic variants as the ones with non-zero genetic dominance effects for rheumatoid arthritis and psoriasis, where the GenoBoost was ranked top as the one with the best predictive performance. We found a modest correlation (Pearson's correlation = 0.70) between the number of genetic variants selected in the GenoBoost PGS model and the fraction of the genetic variants classified as having non-additive genetic dominance effects (Supplementary Fig. 13a, Supplementary Data 1). We also observed modest correlations when we used the number of genetic variants in other PGS models or the estimated additive liability-scale heritability (Supplementary Fig. 13b-f, Supplementary Data 1, 2).

We validated that the inferred mode of genetic inheritance by GenoBoost is highly consistent with what one may infer from the GWAS association statistics conducted under the various genetic inheritance modes[12,14]. Specifically, we conducted GWAS analysis for each genetic variant under additive and all non-additive genetic dominance modes and inferred the genetic inheritance mode to be the one that resulted in the most significant association summary statistics. We found the inferred inheritance modes from the two approaches are largely consistent, although the two approaches rely on different metrics (Fig. 3b).

For example, we found four genetic variants selected by GenoBoost for psoriasis in a 3 M bp intergenic region on chromosome 11 (position: 96M–99M; the nearest transcription start site is at position 96.51 M for the *JRKL* gene). The four genetic variants consist of a mixture of distinct inferred inheritance modes: two genetic variants with additive effects, one with recessive effect, and another with over-dominant effect (Fig. 3c). The inferred inheritance modes for each of the selected genetic variants are highly consistent with the ones from GWAS-based approaches (Fig. 3d). For example, across all of the 900 genetic variants selected in the GenoBoost PGS model for psoriasis, 730 variants (81%) have the same inheritance mode classification. Across the twelve traits, 56%–93% of genetic variants enjoyed agreement with inferred inheritance modes. For 52 of 186 (28%) genetic variants, we found GenoBoost and GWAS classified them into recessive

and additive, respectively, and they consist of the most common discrepancy between the two approaches. The lower number of homozygous careers may result in limited power in recessive GWAS in those cases. On the other hand, GenoBoost considers all individuals with non-missing genotypes when inferring the underlying genetic inheritance mode of the variants, highlighting the advantage of GenoBoost in inferring the underlying inheritance modes of the genetic variants without requiring prior knowledge of the underlying biology.

### Applying GenoBoost to prioritize genetic loci previously not reported in the GWAS catalog

Motivated by the competitive predictive performance and the ability to infer the genetic inheritance modes, we evaluated and subsequently showed the utility of GenoBoost to prioritize genetic loci that were previously not reported in the GWAS catalog[49]. To that end, we identified the list of 240 genetic variants with putative non-additive genetic dominance effects where the relevance of the closest genes (within a 100,000 bp window) to the traits was not previously reported in the GWAS catalog (Fig. 4, Supplementary Fig. 14, Supplementary Data 3, Supplementary Methods). We focused on the three identified variants for rheumatoid arthritis and the top five genetic variants for psoriasis, ranked by their relative contributions in binary classification in GenoBoost for psoriasis, as case studies. We found that four out of the eight genes showed independent support of their relevance to the traits in the literature.

For rheumatoid arthritis, an intronic variant (rs7237982) in the *TNFRSF11A* (*RANK*) gene was prioritized by GenoBoost, with the dominant mode as the inferred mode of inheritance. The *TNFRSF11A* (*RANK*) gene is part of the *RANKL/RANK/OPG* pathway, activated during osteoclast maturation and bone modeling[50]. The *RANK* gene was overexpressed in the rheumatoid arthritis patients compared to control donors[51]. Similarly, the differential expression of the *ARHGAP15* gene, annotated for the closest gene for an intronic variant with dominance (rs2731561), was also reported in the synovial fluid[52].

For psoriasis, an intronic variant (rs7291930) of the *MED15* gene was selected under the inferred dominant mode. The *MED15* gene was overexpressed in psoriatic arthritis patients[53]. Similarly, an intronic variant (rs10193337) for the *SPRED2* gene, which is known to be involved in endochondral ossification for bone development, was also selected by GenoBoost under the over-dominant inheritance mode. The gene is also upregulated in the synovial biopsies in the cases[54]. Beyond the gene body, we found a genetic variant in the *MICD* pseudogene, which was reported to be a potential pleiotropic gene for immune and skeletal disease, including psoriasis[55] (Supplementary Data 3).

Across the eight examples examined, our classification of inheritance mode with GenoBoost is consistent with those from GWAS association summary statistics (Supplementary Figs. 15, 16, Supplementary Table 7). Intriguingly, the associations between rs6773050 and rheumatoid arthritis and rs10193337 and psoriasis were not statistically significant under the additive GWAS analysis, highlighting the advantage of GenoBoost in prioritizing those loci.

### Discussion

We present GenoBoost, a flexible polygenic modeling framework capable of incorporating both additive and non-additive genetic dominance effects without requiring prior knowledge of the genetic architecture of the trait. Systematic benchmarking of GenoBoost against seven commonly used PGS methods demonstrates the competitive predictive performance of GenoBoost, although no single method outperforms the other approaches across the panel of twelve disease outcomes in UK Biobank evaluated in our study. Taking advantage of the sparsity of GenoBoost models, we investigate the factors that may influence the relative predictive performance of PGS methods. We show that GenoBoost improves prediction for

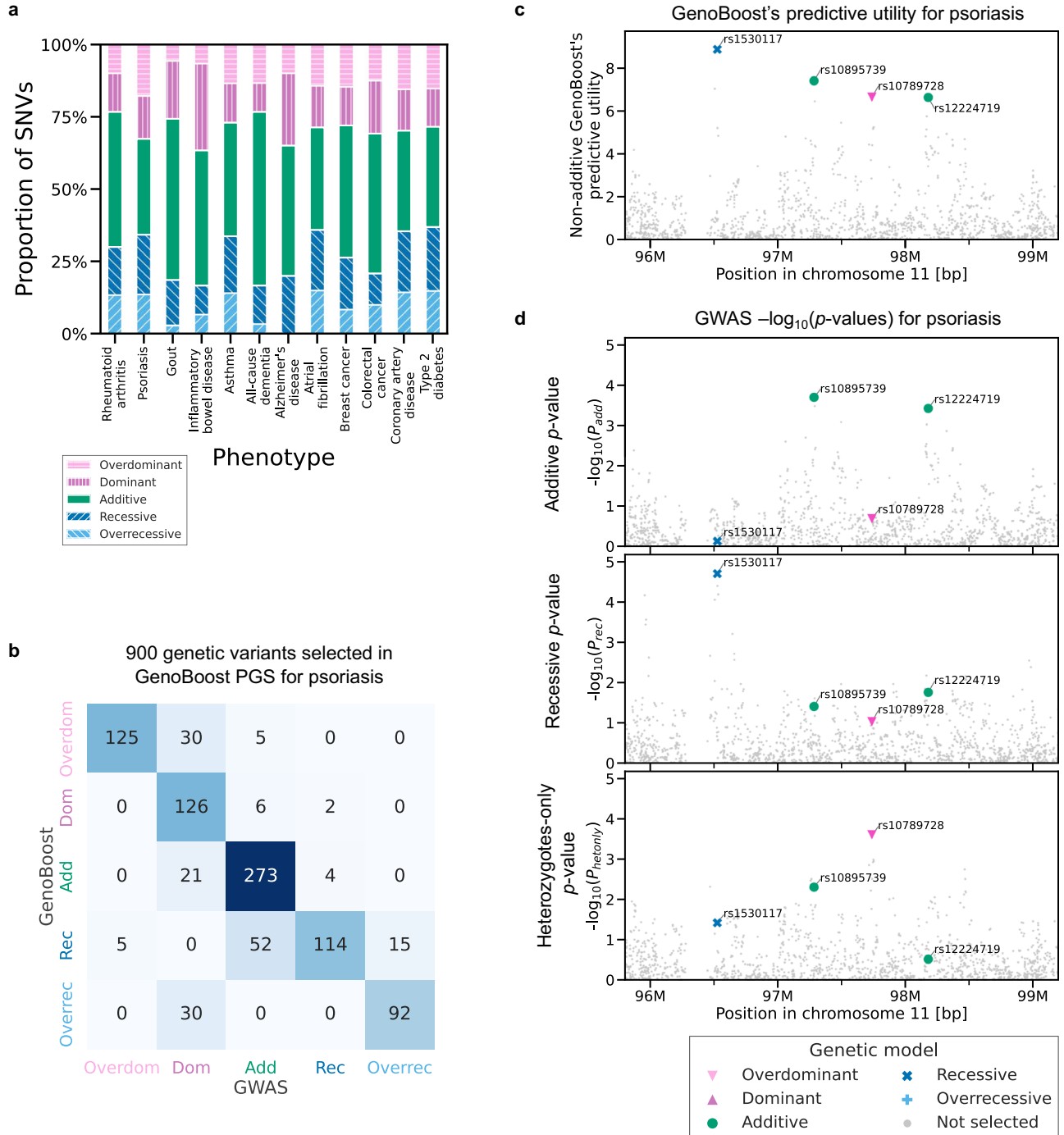

**Fig. 3 | GenoBoost scores allow for the inference of the mode of inheritance.**
**a** For the genetic variants selected in Non-additive GenoBoost models across the twelve disease outcomes in UK Biobank (*x*-axis), the fractions of the inferred inheritance mode are shown (*y*-axis). **b** We compared the inferred inheritance mode from GenoBoost (*y*-axis) and the ones from GWAS summary statistics (*x*-axis) and showed the results as a colored confusion matrix. Add: additive. Dom: dominant. Rec: recessive. **c**, **d** Comparison of GenoBoost scores and GWAS *p*-values, focusing on psoriasis and genetic variants in an intergenic region in chromosome 11. For each genetic variant selected in the Non-additive GenoBoost model within

the 3 Mbp window (*x*-axis), we show the predictive utility of the variant in Non-additive GenoBoost (Supplementary Methods) (**c**). We also show the statistical significance from GWAS for the variant under additive, recessive, and heterozygous-only regression models (**d**). The statistical significance is the nominal *p*-values of the slope of the logistic regression from two-sided tests using up to *n* = 215,768 samples. Four genetic variants with the largest predictive utilities in GenoBoost are highlighted and colored based on the inferred mode of genetic inheritance. Source data are provided as a Source Data file.

autoimmune diseases by incorporating non-additive effects in the MHC locus and, more broadly, works best in less polygenic traits.

We also demonstrate that GenoBoost allows inference of the inheritance mode of each selected genetic variant. We compared the inferred inheritance modes with an orthogonal approach based on

GWAS summary statistics and showed that the inferred modes of inheritance from the two approaches are largely consistent. Focusing on the genetic variants with inferred non-additive effects, we demonstrate the ability of GenoBoost to prioritize genetic loci with genetic dominance that were previously not reported in the GWAS catalog[49].

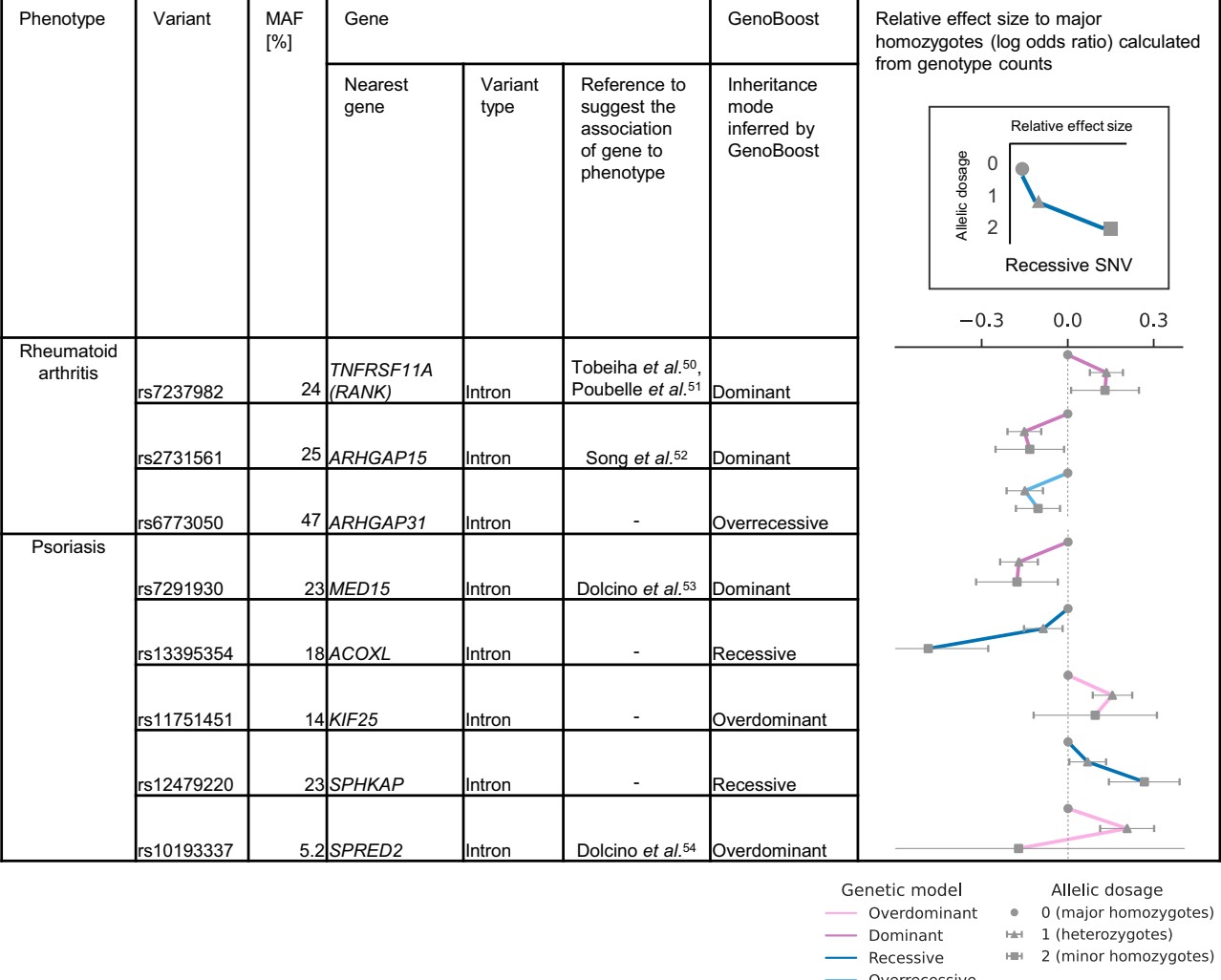

**Fig. 4 | Prioritizing genetic loci previously not reported in the literature with GenoBoost.** Three and five genetic variants selected for rheumatoid arthritis and psoriasis in GenoBoost are shown as illustrative examples, where GenoBoost selected the genetic variants with inferred non-additive genetic dominance effects, and their closest genes (within the 1Mbp window) were not reported in the GWAS catalog. We show the following in the table: phenotype, variant, minor allele frequency (MAF), annotated gene symbol, and predicted consequence of genetic variant, references for the literature reporting the relevance of the closest gene in the phenotype, inferred mode of inheritance. The line plots in the rightmost column represent the log odds ratio (x-axis) of the sample counts of heterozygous and homozygous minor alleles relative to homozygous major alleles. The error bars represent the 95% confidence intervals for Wald's statistics with $n = 215,768$ samples. The three dots connected by a line (y-axis) represent the allelic dosage of 0, 1, and 2 from the top to bottom. The color represents the inferred mode of genetic inheritance by GenoBoost. Source data are provided as a Source Data file.

Methodologically, our study provides substantial advancements in PGS modeling efforts for complex traits. For example, to the best of our knowledge, GenoBoost is the first PGS method that can consider the effects of additive and non-additive genetic dominance effects simultaneously. The advantage of our analytical solution is evident when we compare GenoBoost against snpboost[27], a recently developed PGS method based on the application of statistical boosting on the individual-level genetic data: snpboost is based on $L_2$-boosting[56], a specific kind of statistical boosting technique best suited for quantitative traits, and is limited to additive effects; GenoBoost, in contrast, builds on LogitBoost, which was originally developed for binary classification[33], and also on our theoretical results. The analytical solutions for the optimal GenoBoost scores (Supplementary Methods) overcome the limitations in previous approaches and allow efficient inference of additive and non-additive genetic dominance effects. Indeed, GenoBoost is efficient and is applicable to large-scale datasets from population-based cohorts. In our application of GenoBoost to the UK Biobank dataset consisting of ~1.1 million genetic variants and

$n_{train}$ ~ 216,000 individuals, GenoBoost required only up to 10 h and 30 GB of memory on our cluster computing system consisting of a 32-core CPU (AMD EPYC 7302, 1.5 GHz clock rate) and 2TB of memory (DDR4 RAM, 2.6 GHz clock rate).

Biologically, our flexible and sparse GenoBoost PGS models offer interpretation. We showed the ability of GenoBoost to infer the genetic inheritance mode based on the three GenoBoost scores ($s_0$, $s_1$, and $s_2$). We demonstrated that our classification of inheritance mode was highly consistent with that from GWAS association summary statistics. Compared with the GWAS-based approach, which requires a series of association analyses with varying inheritance mode assumptions, GenoBoost readily inferred genetic inheritance mode while fitting a PGS model in a single run. The ability of GenoBoost to consider all the samples in inferring the mode of genetic inheritance can be advantageous when searching for recessive effects. We further showed the ability of GenoBoost to prioritize genetic loci that were not previously reported in the GWAS catalog[49] by demonstrating that four out of eight tested

genes prioritized by GenoBoost have literature support for their relevance to the trait.

There are several directions for future studies. First, the current implementation of GenoBoost supports binary phenotypes alone; future expansion of the methodology into quantitative traits would be helpful. Second, our current approach operates directly on the individual-level data; further extension of our method coupled with the increasing availability of non-additive GWAS summary statistics would expand the opportunity for joint modeling of additive and non-additive genetic effects across multiple traits and cohorts. Third, our study focused on common SNVs alone; future studies should incorporate a broader type of genetic variants, including indels, HLA allelotypes[57], copy number variations, short tandem repeat expansions, microsatellites, and structural variants, given the increasing catalog of rare genetic variants from short- and long-read sequencing-based studies and methodological innovations of integrating those effects in genetic scores[58]. Fourth, we currently focus on European individuals alone in UK Biobank as proof of principle, but future studies will benefit greatly by incorporating individuals across diverse genetic ancestry from multiple cohorts[25,59]. Fifth, our current model did not consider the interaction between genetic variants; future non-additive PGS models should incorporate such effects.

Our results highlight the benefits of incorporating non-additive genetic dominance effects in PGS models. We demonstrate the advantage of our sparse PGS models in improving predictive performance for autoimmune diseases and less polygenic traits with non-additive effects. We make the coefficients of GenoBoost models publicly available in the PGS catalog (publication ID: PGP000546)[60].

## Methods

### Ethics
This research was conducted using the UK Biobank Resource under Application Number 48405, "Understanding disease subtypes from genotype information" (https://www.ukbiobank.ac.uk/enable-your-research/approved-research/understanding-disease-subtypes-from-genotype-information). All participants of the UK Biobank provided written informed consent (more information is available at https://www.ukbiobank.ac.uk/2018/02/gdpr/).

### The study population in UK Biobank
UK Biobank is a population-based cohort study with genomic and phenotypic datasets across about 500,000 volunteers collected across multiple sites in the United Kingdom[35,36]. We performed sample-level quality control (QC). We focused on unrelated individuals with genetic data based on the following criteria: (1) not reported in "Outliers for heterozygosity or missing rate" (UK Biobank Data Field 22027); (2) not reported in "Sex chromosome aneuploidy" (Data Field 22019); and (3) used to compute principal components (Data Field 22020). Using a combination of self-reported ethnic background (Data Field 21000) and genotype principal components (Data Field 22009), we subsequently defined white British ($n = 337{,}138$), African ($n = 6487$), South Asian ($n = 7952$), and East Asian ($n = 1770$) individuals (Supplementary Methods)[40].

For the 337,138 unrelated white British (WB) individuals, we randomly split them into the model development ($n = 269{,}710$ individuals; equivalent to 80% of the unrelated WB individuals) and the held-out test ($n = 67{,}428$, 20%) sets without using phenotypes (Supplementary Fig. 1). For cross-validation, we further randomly split the model development set into five folds, where we used four of the five folds ($n_{train} = 215{,}768$, 64%, consisting of $n_{train} = 115{,}722$ female samples) and the remaining fold ($n = 53{,}942$, 16%) for model fitting and optimizing the hyperparameters in the model, respectively. We used the held-out test set to evaluate the predictive performance. One of the folds was randomly selected as the primary fold. For breast cancer, we focused

on the female individuals. We used the same population split for all tested traits and PGS methods.

For the 6,487 African, 7,952 South Asian, and 1,770 East Asian individuals, defined using the self-reported ethnic background and genotype principal components (Data Field 22009) (Supplementary Methods), we randomly split them into validation (20%) and test sets (80%). We used the validation set of individuals to account for the covariate effects and the test set to evaluate the predictive performance of PGS models.

### Variant annotation and quality control in UK Biobank
Throughout the study, we used the imputed genotypes (release version 3) and GRCh37 human reference genome. We performed variant annotation with Ensembl's Variant Effect Predictor (VEP) (version 110)[61]. We focused on variants passing the following criteria: (1) unambiguous single nucleotide variants (SNVs) where both reference and alternate alleles are represented by one of the four canonical nucleobases (A, T, G, C); (2) minor allele frequency (MAF) >1%; (3) Hardy-Weinberg disequilibrium test $p$-value $> 1.0 \times 10^{-6}$; (4) the missingness of the variant is <5%; (5) imputation quality score (INFO score) >0.3; and (6) present in the HapMap Phase 3 dataset[62]. We focused on the most and the second most major alleles for multiallelic sites and set the remaining alleles as missing. We computed the quality control metrics using PLINK 2.0[63]. The quality control procedure above resulted in 1,073,718 variants considered in the analysis.

For prioritizing genetic loci with GenoBoost, we annotated the closest gene (up to 100,000 base pairs) for each genetic variant. To select the lead tagging associations in linkage disequilibrium, we excluded variants if another variant with larger predictive utility was already selected within a 1 million base pair window.

### Phenotype definition in UK Biobank
We defined a panel of twelve disease outcomes in UK Biobank by combining the following data sources: (1) self-reported diagnoses, (2) cancer diagnosis code from the UK Cancer Registry, (3) disease diagnoses from the UK National Health Service Hospital Episode Statistics, and (4) operative procedure from the Health Episode Statistics records coded in the Office of Population Censuses and Surveys Classification of Interventions and Procedures, version 4 (OPCS-4) (Table 1, Supplementary Methods). The self-reported diagnoses of cancer and non-cancer outcomes were collected at UK Biobank's assessment center across up to four instances, each of which corresponds to (1) the initial assessment visit (2006-2010), (2) the first repeat assessment visit (2012-2013), (3) the imaging visit (2014-present), and (4) first repeat imaging visit (2019-present). We assigned "case" status if the participants were classified as the case in at least one of the data sources and "control" otherwise.

### Genome-wide association analysis
We applied genome-wide association analysis with PLINK (v2.00 alpha)[63] using age, sex, and the first ten genotype PCs (Data Field 22009) provided by UK Biobank as covariates implemented as "--glm zs omit-ref firth-fallback" command in PLINK2. We normalized covariates using the "--covar-variance-standardize" command. We computed the age of the participants in March 2020 based on year and month of birth (Data Field 33). To benchmark the predictive performance of summary-statistics-based PGS methods trained on the same number of individuals, we applied genome-wide association analysis using the default additive model, focusing on the $n_{train} = 115{,}722$ female individuals (breast cancer) or $n_{train} = 215{,}768$ individuals (the other eleven traits) in the training set. We used the resulting summary statistics for lassosum[6], C + T[4], LDpred[5], PRS-CS[7], and SBayesR[8], as described below. To infer the mode of genetic inheritance using the

statistical significance of GWAS associations, we repeated the GWAS analysis with PLINK using the following list of modifiers in the "--glm" command: "dominant", "recessive", and "hetonly".

## Overview of the GenoBoost algorithm

We fit sparse polygenic score models using GenoBoost applied on the individual-level data consisting of covariate-adjusted phenotypes $y = (y_1, ..., y_n)^T \in R^n$ and genotypic dosage and covariate matrix $X = (x_1, ..., x_n)^T \in R^{n \times (d+c)}$, where $n$ is the number of individuals in the training set, $d$ is the number of genetic variants, and $c$ is the number of covariates. We designed GenoBoost based on statistical boosting[26], and we fit polygenic score models in an iterative procedure, where we gradually expand the set of genetic variants considered in the boosting model (Algorithm 1). To minimize the risks of confounding, we first adjusted for the effects of covariates using logistic regression. In each iteration with GenoBoost, we focus on the residualized phenotypes adjusted for the effects of covariates and genetic variants that have already been selected in the previous iteration steps and identify the most informative genetic variant in predicting the residualized phenotype. Specifically, we defined the predictive utility of each genetic variant as their potential contribution to reducing the objective function (loss function) of GenoBoost (Supplementary Methods). We infer additive and non-additive effects of the selected genetic variants and assign weights for homozygous and heterozygous minor alleles relative to homozygous major alleles using analytical solutions. To minimize the impact of high variance stemming from a small number of minor homozygous carriers, we regularized the genotype-dependent scores for the homozygous minor alleles ($s_2$) so that they will not have extremely large absolute values (Supplementary Methods, Supplementary Fig. 6). We apply batch screening for large-scale datasets, as shown in the full description of the GenoBoost algorithm in Supplementary Methods. We provide the theoretical proof of their optimality of the GenoBoost weights in Theorem 1-7 in the Supplementary Methods. The number of iterations, $T$, and the learning rate, $\gamma$, are hyperparameters we optimize via five-fold cross-validation.

## Algorithm 1. A simplified description of Additive and Non-Additive GenoBoost.

Given: training data $\{(x_1, y_1), (x_2, y_2), \ldots, (x_N, y_N)\}$
User-given parameter: learning rate $0 < \gamma \le 1$ and the maximum iteration count $T$
Initialize: $t = 0$, $F_0(x) = F_{cov}(x)$
For $t = 0, \ldots, T-1$:
1. Compute the probability of sample $i$ being a disease case $p_{t,i}$, the sample working response $z_{t,i}$, and the sample weight $w_{t,i}$:

$$p_{t,i} = \frac{1}{1 + \exp(-F_t(x_i))}, z_{t,i} = \begin{cases} \frac{1}{p_{t,i}} & (y_i = +1) \\ -\frac{1}{1-p_{t,i}} & (y_i = -1) \end{cases}, w_{t,i} = p_{t,i}(1 - p_{t,i}),$$

(1)

and set $W_{t,k} = \sum_{i:G_t(x_i)=k} w_{t,i}$ and $U_{t,k} = \sum_{i:G_t(x_i)=k} w_{t,i} z_{t,i}$.

2. For each SNV, compute the parameters for the optimal GenoBoost scores under the additive model:

$$c_t = \frac{(W_{t,1} + 4W_{t,2})U_{t,0} + 2W_{t,2}U_{t,1} - W_{t,1}U_{t,2}}{W_{t,0}W_{t,1} + W_{t,1}W_{t,2} + 4W_{t,2}W_{t,0}},$$

(2)

$$\alpha_t = \frac{(-W_{t,1} - 2W_{t,2})U_{t,0} + (-W_{t,2} + W_{t,0})U_{t,1} + (2W_{t,0} + W_{t,1})U_{t,2}}{W_{t,0}W_{t,1} + W_{t,1}W_{t,2} + 4W_{t,2}W_{t,0}}.$$

(3)

To consider the non-additive model, set:

$$s_{t,k} = \frac{U_{t,k}}{W_{t,k}}.$$

(4)

3. Compute the loss function $\sum_{i=1}^{N} w_{t,i}(f_t(x_i) - z_{t,i})^2$, where $f_t(x_i)$ is defined as in Eq. (5) for the additive model and as in Eq. (6) for the non-additive model:

$$f_t(x_i) = \begin{cases} c_t & G_t(x_i) = 0 \\ c_t + \alpha_t & G_t(x_i) = 1 \\ c_t + 2\alpha_t & G_t(x_i) = 2 \end{cases},$$

(5)

$$f_t(x_i) = \begin{cases} s_{t,0} & G_t(x_i) = 0 \\ s_{t,1} & G_t(x_i) = 1 \\ s_{t,2} & G_t(x_i) = 2 \end{cases}.$$

(6)

4. Select the SNV with the smallest loss function and update the predictor:

$$F_{t+1}(x) = F_t(x) + \gamma f_t(x).$$

(7)

Output: $F_T(x)$

We considered two variants of GenoBoost models: Additive GenoBoost and Non-additive GenoBoost. When fitting Additive GenoBoost models, we imposed an additional constraint so that non-additive genetic dominance effects are always set to zero (Supplementary Methods). When fitting Non-additive GenoBoost, we did not apply such a constraint, resulting in a model that considered both additive and non-additive effects. Using the validation set metric (covariate-adjusted pseudo-$R^2$), we selected the best-performing model and reported it as the GenoBoost model.

## PGS performance evaluation

We evaluated the predictive performance of PGS models. We evaluated the following four predictive performance metrics in the held-out test set individuals: (1) covariate-adjusted pseudo-$R^2$, (2) odds ratio stratified at the top 1% of the predicted genetic liability, (3) area under the receiver operating characteristic curve (AUC), and (4) area under the precision-recall curve (AUPRC).

**Covariate-adjusted pseudo-$R^2$.** To assess the overall goodness of fit of the PGS models, we evaluated covariate-adjusted pseudo-$R^2$ as in the literature[7], which can be interpreted as Nagelkerke's pseudo-$R^2$ (also known as Cragg and Uhler's pseudo-$R^2$)[37,38] computed for the covariate-adjusted phenotype and defined as

$$R^2 = \left(1 - (L_{covars}/L_{full})^{2/n}\right) / \left(1 - L_{covars}^{2/n}\right),$$

(8)

where $L_{covars}$ and $L_{full}$ represent the likelihood of the covariate-only model and the full model that considers both covariates and genetic variants, respectively, and $n$ represents the sample size. We fit a model on the validation set, obtained the coefficients of covariates and genotype-only models, and reported the predictive performance in the held-out test set.

**Odds ratio.** To assess the ability of PGS models to stratify the individuals with high genetic liability of the disease in a population, we evaluated the odds ratio of disease incidence rates stratified by the top 1% of genotype-only PGS value. Specifically, we focused on the individuals in the held-out test set, ranked them based on the PGS scores, defined high-risk and lower-risk groups as the individuals in the top first percentile and the remainder of the population, and evaluated the

odds ratio using the disease incidence rates in the two stratified groups. We also evaluated the odds ratio at different threshold levels, i.e., the top 3%, 5%, and 10%.

**AUC.** To assess the overall performance of PGS models in risk stratification at different threshold levels, we evaluated the area under the receiver operating characteristic curve (AUC), a commonly used metric for binary classification tasks. AUC values would be 0.5 and 1.0 for random and perfect classifiers, respectively.

**AUPRC.** To assess the ability of PGS in stratifying case individuals in populations, we evaluated the area under the precision-recall curve (AUPRC), another commonly used metric for binary classification for imbalanced datasets. AUPRC values would be equivalent to the proportion of the cases and 1.0 for random and perfect classifiers, respectively.

### Applying GenoBoost to UK Biobank disease outcomes

We applied GenoBoost to a panel of twelve disease outcomes in UK Biobank (Table 1). We considered the effects of the same set of covariate terms as in GWAS, i.e., age, sex, and the first ten genotype PCs. We set the batch size $M_{batch}$ to 50 (Supplementary Methods). We applied five-fold cross-validation and optimized hyperparameters via grid search based on covariate-adjusted pseudo-$R^2$. Specifically, we considered four values (0.05, 0.1, 0.2, and 0.5) for learning rate, $\gamma$, and the following 29 values for the maximum iteration count, $T$: [5, 10, 20, 30, ..., 90, 100, 200, 300, ..., 900, 1000, 2000, 3000,..., 9000, 10000] (Supplementary Fig. 17). The maximum number of unique selected genetic variants is constrained to be 10,000, given the maximum iteration count, $T$, tested in the grid search. We confirmed that the cross-validation selected <10,000 genetic variants (Supplementary Fig. 18, Supplementary Data 1). We reported all of the predictive performance metrics, i.e., covariate-adjusted pseudo-$R^2$, odds ratio for the top 1% individuals, AUC, and AUPRC, evaluated in the held-out test set using the five models from the cross-validation.

### Additional analyses on the MHC locus, polygenicity, and regularization for the homozygous alleles

We performed additional analyses of GenoBoost by excluding chromosome 6 to investigate the effects of the MHC locus. We conducted a simulation study to investigate the effects of polygenicity. We introduced regularization for GenoBoost scores for the homozygous effect alleles. We provide methodological details for those additional analyses in Supplementary Methods.

### Applying previously published PGS methods to UK Biobank disease outcomes

We applied seven previously published PGS methods to the twelve disease outcomes in UK Biobank. We considered five GWAS summary-statistics-based methods (C + T[4], Lassosum[6], LDpred[5], PRS-CS[7], and SBayesR[8]) and additional two methods (snpboost[27] and snpnet[20]) based on the individual-level data. We evaluated the predictive performance of each model and compared them against the ones from GenoBoost. The details of the application of the seven previously published PGS methods are described in the Supplementary Methods.

### Incorporating non-additive effects in snpnet

With snpnet[20], we considered non-additive genetic dominance effects by augmenting its input files for the genetic data. Specifically, we prepared a dummy non-additive genotype matrix $D \in R^{n \times d}$ from the corresponding genotypic dosage matrix $G \in R^{n \times d}$. $D_{i,j} \in \{0, 1\}$ for $j$-th SNV of the $i$-th sample is defined as $D_{i,j} = 1$ if $G_{i,j} = 1$ and $D_{i,j} = 0$ if $G_{i,j} = 0$ or 2. Applying snpnet on $[G, D]$, instead of applying it on $G$, we incorporated non-additive genetic dominance effects in snpnet via learning the polygenic function $G\beta + D\gamma$.

### Inferring mode of genetic inheritance with GenoBoost scores

We used the triplet of GenoBoost scores ($s_0$, $s_1$, and $s_2$) to infer the mode of genetic inheritance for each genetic variant selected in Non-additive GenoBoost models (Supplementary Methods). For the same set of genetic variants, we compared the magnitude of statistical significance of the associations from GWAS under additive and dominance regression models (dominant, recessive, and heterozygotes-only). We selected the genetic inheritance mode that resulted in the most significant association for each variant and considered that as the inferred mode of genetic inheritance with GWAS summary statistics.

### Estimating additive heritability of UK Biobank disease outcomes

We used all samples in the training, validation, and test datasets to perform genome-wide association analysis using PLINK 2.0[63]. We used the GWAS summary statistics to estimate additive heritability in the liability-scale without ascertainment correction using d-ldsc software[15].

### Validation of selected SNVs in the GWAS catalog and literature

We downloaded the GWAS catalog data (version 1.0) for the twelve disease phenotypes considered in the study[49]. We focused on the genetic variants selected in GenoBoost trained on the primary fold and checked whether the locus was previously reported in the GWAS catalog (Supplementary Methods).

### Statistics

For computational and statistical analysis, we used Python. For visualization, we used matplotlib[64] and seaborn[65]. The $p$-values were computed from two-sided tests unless otherwise specified.

### Reporting summary

Further information on research design is available in the Nature Portfolio Reporting Summary linked to this article.

## Data availability

The analyses presented in this study were based on the individual-level data accessed through UK Biobank: https://www.ukbiobank.ac.uk. This research was conducted using the UK Biobank Resource under Application Number 48405. We used the reference panels from the 1000 genomes project (https://www.internationalgenome.org/) and the list of genetic variants from the HapMap3 projects (https://www.broadinstitute.org/medical-and-population-genetics/hapmap-3). The PGS model weights generated from this study are publicly available in the PGS catalog (publication ID: PGP000546). The experimental data generated in this study have been deposited in the Zenodo database under accession code (https://doi.org/10.5281/zenodo.10200754). Source data are provided with this paper.

## Code availability

The GenoBoost software is available on GitHub (https://github.com/rickyota/genoboost) and also as a Docker image (https://hub.docker.com/repository/docker/rickyota/genoboost/). The analysis scripts used in the manuscript are available on GitHub (https://github.com/rickyota/genoboost-paper-script). We also deposit the contents of the GitHub repositories at Zenodo datasets: GenoBoost software (https://doi.org/10.5281/zenodo.10205707) and analysis scripts (https://doi.org/10.5281/zenodo.10200597). The code is released under the GNU General Public License version 3.0.

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

## Acknowledgements

Rikifumi Ohta and Shinichi Morishita would like to thank Riki Kawahara and Hiroaki Hosokawa for carefully reading the proof of the main theorem and for giving valuable comments and suggestions. This research was conducted using the UK Biobank Resource under Application Number 48405. We thank all participants of the UK Biobank. This work uses data provided by patients and collected by the NHS as part of their care and support. Copyright © (2023), NHS England. Re-used with the permission of the NHS England and UK Biobank. All rights reserved. This research used data assets made available by National Safe Haven as part of the Data and Connectivity National Core Study, led by Health Data Research UK in partnership with the Office for National Statistics and funded by UK Research and Innovation (research which commenced between 1st October 2020 – 31st March 2021 grant ref MC_PC_20029; 1st April 2021 -30th September 2022 grant ref MC_PC_20058). This research was funded by Grant-in-Aid for JSPS Fellows 21J21867 (R.O.) and Japan Agency for Medical Research and Development (AMED) 23tm0424219h0003 (S.M.) and was supported in part by National Institutes of Health grants AG054012, AG058002, MH109978, AG062377, AG081017, NS129032, AG077227, NS110453, NS115064, AG062335, AG074003, NS127187, AG067151, MH119509, HG008155, and DA053631 (M.K.). The content is solely the authors' responsibility and does not necessarily represent the official views of the funding agencies; funders had no role in study design, data collection and analysis, the decision to publish, or the preparation of the manuscript.

## Author contributions

These authors contributed equally: R.O. and Y.T.; The corresponding authors are R.O., Y.T., M.K., and S.M.; S.M., Y.T., and Y.S. designed the study; R.O. implemented the software, performed all analyses, and produced the figures with supervision by S.M. and Y.T.; Y.T., S.M., and R.O. wrote the original draft; All authors reviewed and approved the final version of the manuscript; and S.M. and M.K. were responsible for funding acquisition.

## Competing interests

The authors declare no competing interests.
