## [Peer Review File · Nature Communications]

A Polygenic Score Method Boosted by Non-additive ModelsReviewer #1 (Remarks to the Author):

In this work, Ohta and colleagues extended the LogitBoost framework for modelling non-additive genetic effects, in order to improve the predictive performance of polygenic risk scores. The authors applied their method, GenoBoost, to predict risks of 12 diseases in the UK Biobank, and demonstrated improved prediction accuracy for some of these diseases achieved by non-additive effect-inclusive models. The manuscript is clearly written. However, I have a number of concerns, mostly regarding the generalizability of the method and some of the interpretations.

1. First of all, GenoBoost achieved the best predictive performance only when predicting five autoimmune/inflammatory diseases (asthma, gout, inflammatory bowel disease, rheumatoid arthritis, and psoriasis), which are correlated with each other. For other diseases that are more polygenic, including cancer (breast cancer and colorectal cancer), cardiovascular (atrial fibrillation and coronary artery disease), and metabolic (type 2 diabetes) diseases, GenoBoost was consistently outperformed by LDpred, and occasionally PRS-CS. Therefore, I am not convinced that GenoBoost will have broad utility for a wide variety of diseases and am not sure to what extent genetic effect compositions in Figure 3a can be interpreted. Could the authors conduct simulation studies to examine the impacts of (i) disease heritability and (ii) polygenicity (number of causal variants and per-variant effect size) on the performance of GenoBoost?
2. Related to the comment above, as the authors mentioned, MHC region plays an important role in autoimmune/inflammatory diseases. Extended Figures 4 and 5 showed that modelling the MHC region can be challenging for other methods, especially LDpred. Could the authors compare the performance of methods excluding predictors on Chr6, to assess whether the improved prediction accuracy for these diseases was due to better modelling of MHC? The implementation of GenoBoost to avoid extreme SNV values is an important contribution, but may not support the use of this method to predict more polygenic diseases.
3. The authors compared GenoBoost to LASSO using individual-level data. Could a comparison to non-additive effect-inclusive LASSO models be added? That could simply be $Z \sim G + X$, where G is the genotype (0/1/2) and $X = 1$ if $G = 1$; $X = 0$ if $G = 0$ or 2 , which would only double the degree of freedom. This may demonstrate the advantage of having the boosting framework.
4. The disease outcomes in the UK Biobank mostly have a highly imbalanced case-control ratio. The authors should add AUPRC as an additional metric for evaluating model performance.
5. The portability of risk scores created by GenoBoost has not been tested in other cohorts or in non-European ancestry sub-populations in the UK Biobank. Would the inclusion of more complicated genetic effects exacerbate the portability?
6. The limitations that GenoBoost only handles binary outcomes and relies on availability of individual-level data should be discussed.

Reviewer #2 (Remarks to the Author):

Ohta and colleagues propose an innovative method (GenoBoost) for polygenic prediction taking into account both additive and non-additive effects. GenoBoost is based on the boosting framework and the prediction performances were also compared with different alternative PGS approaches showing competitive results and identifying potential candidate variants with non-linear (e.g., dominant and recessive) effect. The manuscript is well written and the analysis can be of high interest in the genetic epidemiology community.

However, the reviewer believe that a few major points should be considered to improve the work.

In particular:

1.Despite comparisons with different PRS methods based on summary statistics are reported, there is only one comparison with a tool based on multi-variable regression directly from genotyping data, namely snpnet which exploits the lasso framework. Since starting from genotyping provide more information with respect to GWAS, I would suggest to include another individual-level genotype PGS model approach, e.g., snpboost tool (<https://github.com/hklinkhammer/snpboost>) which could represent an optimal comparison by being also based on boosting framework to derive sparse additive PGS models.

2.Concerning the missing heritability issue (gap between observed heritability and the one explained by additive SNP-effect) there is an open discussion regarding the potential sources: including non-linear effect, rare-variants role and gene-environment interactions. Recent works suggest that at least for highly polygenic trait (e.g., height) once there is enough data the linear additive models seem to capture the complete heritability due to common variants (<https://www.nature.com/articles/s41586-022-05275-y>). In this regard the reported results in which non-linear GenoBoost significantly outperforms linear GenoBoost (i.e., rheumatoid arthritis and psoriasis) would be of high interest. Could authors check if there is an (inverse) correlation between the polygenicity of the signal (e.g., number of selected variants) and the proportion of non-linear detected variants. This would align with the current literature suggesting that highly polygenic traits are mainly characterized by additive effects at the level of common variants whereas low polygenic traits can be characterized by a sparse genetic architecture including also not-negligible non-linear effects.

3.For the validation of the detected variants with non-linear effects authors considered the closest genes and evaluated gene-phenotype associations from literatures. This is definitely a valid approach to corroborate the findings, however, I would suggest given the availability of genotype/phenotype data to test dominant and recessive models via univariable association directly on the genotyping data. This would allow a direct comparison of the selected PGS variant, in fact, the variant-gene match especially in non-coding regions might be quite complex in terms of biological interpretation (e.g., one could consider also eQTL and other regulatory mechanisms).

4. There are a few traits in which GenoBoost performances are clearly lower than the best alternative performing method (e.g., coronary artery disease and type 2 diabetes). Authors claims this is due to the fact that the alternative Bayesian approaches by incorporating the LD can better compute the heritability of the regions. Would it be possible to analyze and report how the model sparsity scale with respect to the prediction performance (e.g., by evaluating the different learning rate grid results) to check that indeed these are the cases in which the prediction performances have not yet reached a maximum (suggesting that such traits are highly polygenic and thus sparse models are not optimal in terms of prediction in comparison to "omnigenic" models).

Minors:

- I would suggest to rephrase the 1st sentence in the abstract:

I would skip the word "improve" as there is no comparison but rather just indicating that "PGS can be used for phenotype prediction by aggregating..."

similarly, I would avoid the term "however" as there is no contrast but rather directly report the non-linear effects are typically neglected in PGS models.

- The acronym SNV was not initially defined

- I would remodulate a little some sentences referring to the relevance of non-linear effects and GenoBoost:

e.g. "Our results demonstrate that the non-additive model is essential in predicting risk for polygenic diseases."

For some of the reported phenotypes that is not what can be observed in the results, I would rather

write that according to genetic architecture of the underlying traits non-linear models can improve prediction and understanding of the genetic liability.

"we believe that GenoBoost will be an indispensable approach for settling the "missing heritability" problem"

I would instead indicate that GenoBoost can be a useful tool to investigate if the missing heritability of a traits can be attributed the presence of a non-linear component in the genetic liability

Reviewer #3 (Remarks to the Author):

The authors proposed a novel PRS method that integrates both additive and non-additive genetic effects using the boosting algorithm. In the UKB data application, for certain traits that exhibit non-additive genetic architecture, the method showed improved performance as compared to existing PRS approaches.

The idea to incorporate non-additive effects to PRS is novel and optimization through boosting, which has been trending in PRS method research (PMIDs: 28979001, 36704342, 35995843), is a clever approach to combine multiple weak classifiers as often is the case for GWAS signals. However, I find the conclusion that "the non-additive model is essential in predicting risk for polygenic diseases" not founded as non-additive genetic variance contribute very little to complex traits at large as demonstrated by many others (Palmer et al., 2023 - 36996212, Hivert et al., 2021 - 33811805). I do believe that non-additive model is important to study but have doubts about whether the proposed PRS can achieve what the authors suggested, in replacing other additive PRSs that have better performance on complex polygenic traits that have sizeable genetic effects. The authors should focus on defining for which traits their method is expected to excel; the results are already there, but the message is not clear yet. For example, contrast the good results (RA, Asthma, etc) with the bad (T2D, CAD, etc) and really hone in on what makes their method work and otherwise.

Specific comments:

- In the abstract, I find it misleading to say that 40-67% of variants were non-additive in the GenoBoost model since only a small # of SNVs were present in the model to start with. This is obviously one limitation of the method, setting the maximum number of SNVs to be 10,000. Have the authors investigated whether increasing this # would improve performance for traits where it currently underperforms? More specifically, how much of the small # of SNVs included in the final model was driven by Boosting and how much by the underlying genetic architecture of the trait?
- If GenoBoost also considers additive models, then why does it still underperform additive models like lassosum and LDpred? On the other hand, why is the additive GenoBoost better than the other additive models for certain traits?
- As for its application to characterize genetic model, the idea to use PRS to explore whether non-additive variance vs. additive variance is the primary contributor is interesting. But I have a hard time agreeing that the effects estimated from GenoBoost can be considered association effects (line 173-174). These optimized effects were for prediction and should be followed up by a formal testing if any claims were to be made. In any case, the strength of the method is through the aggregated non-additive genetic effects, testing individual SNPs requires sample size much larger as pointed out by Palmer et al. (2023).
- It is not clear how the authors selected these 12 traits and I don't think these are "representative", at least in comparison to what were typically used in the large collection of PRS methods papers.
- The authors need to tone down the conclusion and be careful of the wording. For example, GenoBoost cannot "classify" genetic model (line 203), it simply selects the model that improves prediction. And the authors have not shown it to be "essential" in predicting polygenic disease in which only 5 of the 12 were best.
- Extreme SNV scores: this winsorizing of the effect was to deal with the current sample size, can the authors comment on the choice of the adjustment and how different choices of adjustment value

impact the fit/final results?

- Finally, correction modelling of LD or models that are robust to LD misspecification have been shown to be the key in improving PRSs, thus, I find the discussion on why GenoBoost is unable or how to incorporate in future research too brief. For example the authors might want to address why in some cases (e.g. RA), ignoring LD seems to be okay but in others this has the consequence of worse performance as compared to other methods.

Minor comments:

- Acronyms spell out the first time when mentioning, e.g. GWAS, SNV, etc.
- sBayes is missing for a few traits (Psoriasis, AD) in Figure 1.
- The light pink color in Figure 3-C and D was hard to see.
- Figure 7-8 really difficult to read, also maybe consider collapsing; these p-values are really not that different - and neither passed threshold for significant testing, maybe shown in a table instead?

We would like to thank the reviewers for giving us many valuable comments and suggestions. Please find answers to all comments one by one below.

Reviewer #1 (Remarks to the Author):

Comment: In this work, Ohta and colleagues extended the LogitBoost framework for modelling non-additive genetic effects, in order to improve the predictive performance of polygenic risk scores. The authors applied their method, GenoBoost, to predict risks of 12 diseases in the UK Biobank, and demonstrated improved prediction accuracy for some of these diseases achieved by non-additive effect-inclusive models. The manuscript is clearly written. However, I have a number of concerns, mostly regarding the generalizability of the method and some of the interpretations.

Answer: Thank you very much for your constructive comments and suggestions. We have revised the manuscript accordingly in what follows.

Comment: First of all, GenoBoost achieved the best predictive performance only when predicting five autoimmune/inflammatory diseases (asthma, gout, inflammatory bowel disease, rheumatoid arthritis, and psoriasis), which are correlated with each other.

Answer: Actually, the five autoimmune/inflammatory diseases (asthma, gout, inflammatory bowel disease, rheumatoid arthritis, and psoriasis) are not correlated because the Jaccard similarity of each pair of the five disorders is small (< 0.04) as shown in Supplementary Fig. 18 and Supplementary Notes.

Comment: For other diseases that are more polygenic, including cancer (breast cancer and colorectal cancer), cardiovascular (atrial fibrillation and coronary artery disease), and metabolic (type 2 diabetes) diseases, GenoBoost was consistently outperformed by LDpred, and occasionally PRS-CS. Therefore, I am not convinced that GenoBoost will have broad utility for a wide variety of diseases and am not sure to what extent genetic effect compositions in Figure 3a can be interpreted. Could the authors conduct simulation studies to examine the impacts of (i) disease heritability and (ii) polygenicity (number of causal variants and per-variant effect size) on the performance of GenoBoost?

Answer: We would like to thank the reviewer for valuable suggestion. Accordingly, we designed a simulation dataset with parameters; namely, additive heritability, non-additive (dominance) heritability, and a number of causal variants (see Supplementary Methods). We generated different datasets by specifying different values for the

parameters, and applied GenoBoost and LDpred to the datasets to examine the impacts of the parameters on the two methods. As shown in Supplementary Fig. 11 and Supplementary Notes, in terms of Nagelkerke's R^2 , GeneBoost outperformed LDpred; however, as the number of causal variants increases, the difference becomes smaller, which is consistent with the fact that LDpred performed better than GenoBoost for highly polygenic diseases (e.g., atrial fibrillation, coronary artery disease, breast cancer and colorectal cancer). As expected, non-additive GenoBoost had better accuracy in the presence of dominant heritability, and was comparable to additive GenoBoost otherwise.

Comment: Related to the comment above, as the authors mentioned, MHC region plays an important role in autoimmune/inflammatory diseases. Extended Figures 4 and 5 showed that modelling the MHC region can be challenging for other methods, especially LDpred. Could the authors compare the performance of methods excluding predictors on Chr6, to assess whether the improved prediction accuracy for these diseases was due to better modelling of MHC?

Answer: Thank you for your insightful comment. When trained with whole genotype and excluding Chr6 in the test dataset, GenoBoost ranks first for three out of five phenotypes as shown in Supplementary Fig. 12a and Supplementary Notes. This suggests that GenoBoost successfully captures non-additive effects of MHC region. Additionally, we excluded Chromosome 6 from the training dataset and trained in GenoBoost and LDpred for rheumatoid arthritis and psoriasis, and observed results similar to excluding Chromosome 6 in the test dataset (Supplementary Fig. 12b).

Comment: The implementation of GenoBoost to avoid extreme SNV values is an important contribution, but may not support the use of this method to predict more polygenic diseases.

Answer: When the adjustment is not used, accuracy deteriorated as shown in Supplementary Fig. 17. Accuracy was lower for more polygenic diseases. We mentioned this point in Supplementary Methods.

Comment: The authors compared GenoBoost to LASSO using individual-level data. Could a comparison to non-additive effect-inclusive LASSO models be added? That could simply be $Z \sim G + X$, where G is the genotype (0/1/2) and $X = 1$ if $G = 1$; $X = 0$ if $G = 0$ or 2 , which would only double the degree of freedom. This may demonstrate the advantage of having the boosting framework.

Answer: Thank you for remarking this point. We implemented the non-additive version of LASSO. The non-additive LASSO improved the accuracy for psoriasis, but did not outperform GenoBoost (Supplementary Fig. 13, Supplementary Notes). The code script is available on GitHub (<https://github.com/rickyota/genoboost-paper-script>).

Comment: The disease outcomes in the UK Biobank mostly have a highly imbalanced case-control ratio. The authors should add AUPRC as an additional metric for evaluating model performance.

Answer: Accordingly, AUPRC is also used in addition to Nagelkerke's R^2 and AUC so as to evaluate the predictive accuracy of each method (see Supplementary Fig. 2c). In general, the relative accuracy between methods did not change for all diseases except IBD. One reason for the increased AUPRC in IBD might be that the case-control ratio for IBD is 1.1%, which is relatively small compared to other diseases, though GenoBoost has a high AUPRC for Alzheimer's disease with a case-control rate of 0.5%.

Comment: The portability of risk scores created by GenoBoost has not been tested in other cohorts or in non-European ancestry sub-populations in the UK Biobank. Would the inclusion of more complicated genetic effects exacerbate the portability?

Answer: We compared the accuracy for the non-European ancestry population in the UK Biobank. The accuracy for the non-European ancestry population was lower than that of white British samples for most methods and phenotypes (Supplementary Fig. 10 and Supplementary Notes). Nagelkerke's R^2 in non-European ancestry tends to be similar to R^2 in white British samples.

Comment: The limitations that GenoBoost only handles binary outcomes and relies on availability of individual-level data should be discussed.

Answer: We described this limitation of GenoBoost in the discussion section as follows: "GenoBoost handles binary outcomes and relies on availability of individual-level data, which is a limitation that should be addressed in future work."

Reviewer #2 (Remarks to the Author):

Comment: Ohta and colleagues propose an innovative method (GenoBoost) for polygenic prediction taking into account both additive and non-additive effects. GenoBoost is based

on the boosting framework and the prediction performances were also compared with different alternative PGS approaches showing competitive results and identifying potential candidate variants with non-linear (e.g., dominant and recessive) effect. The manuscript is well written and the analysis can be of high interest in the genetic epidemiology community.

Answer: Thank you very much for your valuable comments and suggestions. We have revised the manuscript accordingly in what follows.

Comment: However, the reviewer believe that a few major points should be considered to improve the work. In particular: Despite comparisons with different PRS methods based on summary statistics are reported, there is only one comparison with a tool based on multi-variable regression directly from genotyping data, namely snpnet which exploits the lasso framework. Since starting from genotyping provide more information with respect to GWAS, I would suggest to include another individual-level genotype PGS model approach, e.g., snpboost tool (<https://github.com/hklinkhammer/snpboost>) which could represent an optimal comparison by being also based on boosting framework to derive sparse additive PGS models.

Answer: We applied snpboost to the datasets of the 12 diseases, and observed that snpboost was slightly better than GenoBoost and ranked first for asthma, and GenoBoost ranked first for four traits and second for three (Fig. 2, Supplementary Figs. 2-6). We revised the manuscript accordingly.

Comment: Concerning the missing heritability issue (gap between observed heritability and the one explained by additive SNP-effect) there is an open discussion regarding the potential sources: including non-linear effect, rare-variants role and gene-environment interactions. Recent works suggest that at least for highly polygenic trait (e.g., height) once there is enough data the linear additive models seem to capture the complete heritability due to common variants (<https://www.nature.com/articles/s41586-022-05275-y>). In this regard the reported results in which non-linear GenoBoost significantly outperforms linear GenoBoost (i.e., rheumatoid arthritis and psoriasis) would be of high interest.

Answer: Thank you for pointing out these points. We read the paper Yengo *et al.*, 2022 and confirmed that the SNP heritability estimated by was under the additive model only, and the non-additive (dominance) model was not considered at all. Meanwhile, according to Palmer *et al.*, 2023, non-linear effects exist for some phenotypes.

Comment: Could authors check if there is an (inverse) correlation between the polygenicity of the signal (e.g., number of selected variants) and the proportion of non-linear detected variants. This would align with the current literature suggesting that highly polygenic traits are mainly characterized by additive effects at the level of common variants whereas low polygenic traits can be characterized by a sparse genetic architecture including also not-negligible non-linear effects.

Answer: Although this is an important speculation, we found no correlation between the polygenicity of the signal (e.g., number of selected variants) and the proportion of non-linear detected variants (Supplementary Fig. 19, Supplementary Notes).

Comment: For the validation of the detected variants with non-linear effects authors considered the closest genes and evaluated gene-phenotype associations from literatures. This is definitely a valid approach to corroborate the findings, however, I would suggest given the availability of genotype/phenotype data to test dominant and recessive models via univariable association directly on the genotyping data. This would allow a direct comparison of the selected PGS variant, in fact, the variant-gene match especially in non-coding regions might be quite complex in terms of biological interpretation (e.g., one could consider also eQTL and other regulatory mechanisms).

Answer: We already conducted the univariable association test under the dominant, recessive and heterozygous-only model. The result is shown in Supplementary Figs. 8-9. We investigated whether the selected SNVs were present in other than protein-coding genes. We found several SNVs in long intergenic non-coding RNA but they were not reported for psoriasis. We found an SNV in *MICD* pseudogene, which were reported to be potential pleiotropic genes for immune and skeletal disease including psoriasis in: He, P. *et al.* (2021) 'Identification of Potential Pleiotropic Genes for Immune and Skeletal Diseases Using Multivariate MetaCCA Analysis', *Current genomics*, 22(8), pp. 596–606. We added this point in the discussion section.

Comment: There are a few traits in which GenoBoost performances are clearly lower than the best alternative performing method (e.g., coronary artery disease and type 2 diabetes). Authors claims this is due to the fact that the alternative Bayesian approaches by incorporating the LD can better compute the heritability of the regions. Would it be possible to analyze and report how the model sparsity scale with respect to the prediction performance (e.g., by evaluating the different learning rate grid results) to check that

indeed these are the cases in which the prediction performances have not yet reached a maximum (suggesting that such traits are highly polygenic and thus sparse models are not optimal in terms of prediction in comparison to “omnigenic” models).

Answer: Accordingly, we considered this point and confirmed that the prediction performance reached the maximum (Supplementary Fig. 15).

Comment: I would suggest to rephrase the 1st sentence in the abstract: I would skip the word “improve” as there is no comparison but rather just indicating that “PGS can be used for phenotype prediction by aggregating...” similarly, I would avoid the term “however” as there is no contrast but rather directly report the non-linear effects are typically neglected in PGS models.

Answer: We have revised the sentences as you pointed out.

“Polygenic scores (PGSs) can be used for phenotypic prediction by aggregating the effects of genetic variants; most PGS methods focus only on their additive effects.”

Comment: The acronym SNV was not initially defined

Answer: We describe “single nucleotide variants (SNVs)” accordingly.

Comment: I would remodulate a little some sentences referring to the relevance of non-linear effects and GenoBoost: e.g. “Our results demonstrate that the non-additive model is essential in predicting risk for polygenic diseases.” For some of the reported phenotypes that is not what can be observed in the results, I would rather write that according to genetic architecture of the underlying traits non-linear models can improve prediction and understanding of the genetic liability. “we believe that GenoBoost will be an indispensable approach for settling the “missing heritability” problem” I would instead indicate that GenoBoost can be a useful tool to investigate if the missing heritability of a traits can be attributed the presence of a non-linear component in the genetic liability

Answer: We would like to thank the reviewer for these comments, and accordingly we revised the two sentences as follows:

- “Our results demonstrate that the non-additive models can improve risk prediction for polygenic diseases.”
- “GenoBoost is a useful tool to investigate if the missing heritability of a trait can be attributed to the presence of a non-linear component in the genetic liability.”

Reviewer #3 (Remarks to the Author):

Comment: The authors proposed a novel PRS method that integrates both additive and non-additive genetic effects using the boosting algorithm. In the UKB data application, for certain traits that exhibit non-additive genetic architecture, the method showed improved performance as compared to existing PRS approaches. (1) The idea to incorporate non-additive effects to PRS is novel and optimization through boosting, which has been trending in PRS method research (PMIDs: 28979001, 36704342, 35995843), is a clever approach to combine multiple weak classifiers as often is the case for GWAS signals. However, I find the conclusion that “the non-additive model is essential in predicting risk for polygenic diseases” not founded as non-additive genetic variance contribute very little to complex traits at large as demonstrated by many others (Palmer et al., 2023 - 36996212, Hivert et al., 2021 - 33811805). I do believe that non-additive model is important to study but have doubts about whether the proposed PRS can achieve what the authors suggested, in replacing other additive PRSs that have better performance on complex polygenic traits that have sizeable genetic effects. The authors should focus on defining for which traits their method is expected to excel; the results are already there, but the message is not clear yet. For example, contrast the good results (RA, Asthma, etc) with the bad (T2D, CAD, etc) and really hone in on what makes their method work and otherwise.

Answer: Thank you for the valuable comments. We cited the suggested papers as follows: “Several PGS methods using boosting technique has been proposed. However, none of these were designed to incorporate non-additive effects.”

We also cited the papers referring to non-additive heritability as follows: “Although the non-additive heritability was estimated to be negligible overall, Palmer analyzed non-additive genetic effects on polygenic traits in the UK Biobank and suggested a modest number of loci significantly associated with phenotypes”.

GenoBoost tended to have large accuracy for less polygenic diseases, but asthma was exception. We modified the following sentence to clarify this point:

“Bayesian-based methods can capture the heritability in the region from all of the SNVs by modeling the linkage disequilibrium for highly polygenic phenotypes.”

Comment: Specific comments:(2)- In the abstract, I find it misleading to say that 40-67%

of variants were non-additive in the GenoBoost model since only a small # of SNVs were present in the model to start with. This is obviously one limitation of the method, setting the maximum number of SNVs to be 10,000. Have the authors investigated whether increasing this # would improve performance for traits where it currently underperforms? More specifically, how much of the small # of SNVs included in the final model was driven by Boosting and how much by the underlying genetic architecture of the trait?

Answer: We added the prediction performance on various parameters such as the number of SNVs and learning rates, and we confirmed that the prediction performance reached the maximum (Supplementary Fig. 15). GenoBoost consistently selected fewer SNVs than other methods as shown in Supplementary Fig. 4.

Comment: (3)- If GenoBoost also considers additive models, then why does it still underperform additive models like lassosum and LDpred? On the other hand, why is the additive GenoBoost better than the other additive models for certain traits?

Answer: These are important points that need to be addressed in future research. We described possibilities for further research in the discussion. For highly polygenic diseases except for asthma, GenoBoost is outperformed by LDpred and lassosum presumably because LDpred and lassosum consider a much larger number of SNVs than GenoBoost and is able to capture the heritability in the region from all of the SNVs by modeling the linkage disequilibrium. Therefore, we discussed that the algorithm could be improved by incorporating LD information in the future. For the other less polygenic diseases, additive GenoBoost seems to be able to utilize a small number of representative SNVs from LD implicitly and reduces noise of effect sizes and avoid overfitting.

Comment: (4)- As for its application to characterize genetic model, the idea to use PRS to explore whether non-additive variance vs. additive variance is the primary contributor is interesting. But I have a hard time agreeing that the effects estimated from GenoBoost can be considered association effects (line 173-174). These optimized effects were for prediction and should be followed up by a formal testing if any claims were to be made. In any case, the strength of the method is through the aggregated non-additive genetic effects, testing individual SNPs requires sample size much larger as pointed out by Palmer et al. (2023).

Answer: The optimized effects are almost identical to GWAS association effects because GenoBoost iteratively selects a new SNV that is independent from previously chosen

SNVs. As you pointed out, the strength of GenoBoost is its ability to aggregate non-additive genetic effects unlike non-additive association tests. We emphasized this point in the Discussion section as follows:

“GenoBoost's strength is its ability to aggregate these non-additive genetic effects.”

Comment: (5)- It is not clear how the authors selected these 12 traits and I don't think these are “representative”, at least in comparison to what were typically used in the large collection of PRS methods papers.

Answer: We removed the word “representative” from the sentence.

Comment: (6)- The authors need to tone down the conclusion and be careful of the wording. For example, GenoBoost cannot “classify” genetic model (line 203), it simply selects the model that improves prediction. And the authors have not shown it to be “essential” in predicting polygenic disease in which only 5 of the 12 were best.

Answer: Thank you very much for reading the manuscript carefully. Accordingly, we revised the sentence to read: “Non-additive GenoBoost has the ability to select the model to improve prediction and is useful in prioritizing non-additive variants relevant with polygenic diseases.”

We also revised a sentence in the abstract by: “Our results demonstrate that the non-additive models can improve risk prediction for polygenic diseases.”

Comment: (7)- Extreme SNV scores: this winsorizing of the effect was to deal with the current sample size, can the authors comment on the choice of the adjustment and how different choices of adjustment value impact the fit/final results?

Answer: Indeed, the winsorizing effect size deals with too large s_2 due to a small number of minor heterozygotes. When we did not use the adjustment or used another adjustment (Supplementary Methods), the accuracy dropped largely as shown in Supplementary Fig. 17. We mentioned this point in Supplementary Methods.

Comment: (8)- Finally, correction modelling of LD or models that are robust to LD misspecification have been shown to be the key in improving PRSs, thus, I find the discussion on why GenoBoost is unable or how to incorporate in future research too brief. For example the authors might want to address why in some cases (e.g. RA), ignoring LD seems to be okay but in others this has the consequence of worse performance as compared to other methods.

Answer: In fact, GenoBoost can implicitly capture the heritability of LD by selecting representative SNVs in LD regions. Indeed, several SNVs are typically selected from one LD for all phenotypes including RA. However, explicitly incorporating the LD distribution into GenoBoost may be effective in improving accuracy for some diseases. We described this research direction in the discussion section.

Comment: Acronyms spell out the first time when mentioning, e.g. GWAS, SNV, etc.

Answer: Those acronyms are spelled out in the revised version.

Comment: sBayes is missing for a few traits (Psoriasis, AD) in Figure 1.

Answer: SBayesR did not work for those traits due to convergence error as written in Online Method.

Comment: The light pink color in Figure 3-C and D was hard to see.

Answer: We revised to use dark pink color for better readability.

Comment: Sup. Figure 7-8 really difficult to read, also maybe consider collapsing; these p-values are really not that different - and neither passed threshold for significant testing, maybe shown in a table instead?

Answer: We generated tables for these figures accordingly as Supplementary Table 1.

Reviewer #1 (Remarks to the Author):

The authors have addressed my comments.

Reviewer #2 (Remarks to the Author):

I thank the authors for thoroughly addressing all of my concerns. I believe that the additional analysis and interpretation have significantly improved the manuscript's quality. I believe this work represents a valuable contribution to the field of statistical genetics in the context of polygenic risk modeling. I have no further issues to raise.

Reviewer #3 (Remarks to the Author):

I am overall satisfied with the responses. But there are some remaining gaps that the authors can close to help readers. To contextualize the results, the authors should perhaps stress throughout what is so characteristic about the 12 traits for which genoBoost is competitive with respect to other PRS methods. For starters, it should be clear in the abstract that these are 12 binary phenotypes. Is there any criteria for the selection process that can offer insights? Or specifically for which disease categories that genoBoost seems to offer improved performance, in terms of genetic architecture from existing GWASs.

We are grateful to the reviewers for giving us valuable comments and suggestions. Please find answers to all comments one by one below.

Comment: Reviewer #1: The authors have addressed my comments.

Answer: We would like to thank you for the valuable comments we have received so far.

Comment: Reviewer #2: I thank the authors for thoroughly addressing all of my concerns. I believe that the additional analysis and interpretation have significantly improved the manuscript's quality. I believe this work represents a valuable contribution to the field of statistical genetics in the context of polygenic risk modeling. I have no further issues to raise.

Answer: We would like to thank you for the valuable comments we have received so far.

Comment: Reviewer #3: I am overall satisfied with the responses. But there are some remaining gaps that the authors can close to help readers. To contextualize the results, the authors should perhaps stress throughout what is so characteristic about the 12 traits for which genoBoost is competitive with respect to other PRS methods. Is there any criteria for the selection process that can offer insights? Or specifically for which disease categories that genoBoost seems to offer improved performance, in terms of genetic architecture from existing GWASs.

Answer: Thank you for the valuable comments for improving our manuscript. To clarify the above points, we put the following statement in the abstract:

“We reveal that GenoBoost improves prediction for autoimmune diseases by incorporating non-additive effects localized in the MHC locus and, more broadly, works best in less polygenic traits. GenoBoost finds non-zero genetic dominance effects for 602 of 900 selected genetic variants, resulting in 7.9% improvements in performance for rheumatoid arthritis.”

To further support this statement in more detail, we have included two paragraphs in the results. At the end of the discussion, we added the following conclusion:

“Our results highlight the benefits of incorporating non-additive effects in PGS models. We demonstrate the advantage of our sparse PGS models in improving predictive performance for autoimmune disorders and less polygenic traits with non-additive effects.”

Comment: Reviewer #3: For starters, it should be clear in the abstract that these are 12 binary phenotypes.

Answer: Accordingly, in the abstract, we described that we applied our method to “binary” phenotypes.

Author response to editorial and reviewer comments for manuscript NCOMMS-23-28715C titled “**A Polygenic Score Method Boosted by Non-additive Models**” by Ohta, *et al.*

We are grateful for the very useful comments and suggestions for the last several rounds of the revision, which we believe have helped improve our manuscript. After the last revision, we came to realize that there were critical errors in our previous responses to reviewers’ comments and in earlier versions of the manuscript. After consultation with the handling editor, the editorial team kindly agreed to have another round of revision to resolve scientific inaccuracies in the manuscript and incorporate necessary changes that were inadvertently omitted from the previous revision of the manuscript. We are grateful for the opportunity and confident that the revised manuscript helped us ensure that the presentation of our work meets the transparency and reproducibility standards of the field.

We provide the following items for your review in the current revision package:

1. Description of the major changes necessary to resolve scientific inaccuracy in the manuscript;
2. Point-by-point response to all of the reviewers’ comments to ensure our revised manuscript fully addresses insightful comments and helpful suggestions; and
3. Manually annotated description of changes comparing the previously submitted version (NCOMMS-23-28715B, i.e. the previous submission) and the revised text (NCOMMS-23-28715C, i.e. the current manuscript).

Overall, the revisions have helped us resolve the scientific inaccuracy and clarity of the manuscript, thus making it accessible to a broader range of readers, including medical scientists, suitable for consideration at *Nature Communications*.

Description of the major changes necessary to resolve scientific inaccuracy:

Here we summarize the three major changes to resolve scientific inaccuracy in the last version of the manuscript:

1. Portability of PGS for Non-European samples

Reviewer #1 (round 1, point 5) asked us to examine the portability of polygenic scores trained in white-British samples to non-European ancestry sub-populations. However, in our previous response, we confused the non-European samples with non-British samples, and the results were uninterpretable.

To fix the error, we prepared African, South Asian, and East Asian populations as non-European ancestry sub-populations in the UKBiobank (**Supplementary Methods**). We evaluated the predictive performance (**Supplementary Fig. 12**) and summarized the findings in the main text (quoted below).

Supplementary Figure 12. Prediction accuracy for African, South Asian, and East Asian samples compared to white British samples in UK Biobank. Covariate-adjusted pseudo- R^2 of PGS models across GenoBoost (i, A, B) and seven other methods (ii-viii) for twelve phenotypes for African (a, $n=5,190$), South Asian (b, $n=6,362$), and East Asian (c, $n=1,416$) test samples compared to white British samples were shown. Female samples were used for breast cancer ($n=2,975$; African, $n=2,867$; South Asian, $n=904$). Psoriasis, inflammatory bowel

disease, all-cause dementia, and Alzheimer's disease for East Asian samples are not shown since regression in validation dataset failed due to small number of case samples. On each box, the center line is the median, the top and bottom of the box are the second and fourth value (Q3 and Q1), the upper and lower whiskers are shown if the first and the fifth are in $Q3 + 1.5 \text{ IQR}$ and $Q1 - 1.5 \text{ IQR}$, respectively, where $\text{IQR} = Q3 - Q1$. Source data are provided as a Source Data file.

Results (page 5, lines 226):

We observed a limited predictive performance of GenoBoost in non-European population groups in UK Biobank, as in other PGS methods trained only on genetic datasets from European individuals (Supplementary Fig 12, Supplementary Note 4).

2. Correlation between the polygenicity of the signal and nonlinear variants

Reviewer #2 (round 1, point 2) asked us to check if there is a correlation between the polygenicity of the signal and the fraction of genetic variants with estimated non-zero genetic dominance effects, and we previously answered that there was no correlation between the polygenicity of the signal and the proportion of nonlinear detected variants (quoted below).

Although this is an important speculation, we found no correlation between the polygenicity of the signal (e.g., number of selected variants) and the proportion of nonlinear detected variants (Supplementary Fig. 19, Supplementary Notes).

However, we found that the statement was not accurate. There was a modest correlation (Pearson's correlation of 0.67) between the number of genetic variants (approximate measure of polygenicity) and the fraction of the genetic variants (**Supplementary Fig. 13**). We fixed the results (quoted below).

Results (page 5, lines 240):

We found a modest correlation (Pearson's correlation = 0.67) between the number of genetic variants selected in the GenoBoost PGS model and the fraction of the genetic variants classified as having non-additive genetic dominance effects (Supplementary Fig 13).

3. Improving the clarity of the manuscript

Several reviewers asked us to eliminate inappropriate expressions. However, our previous response did not fully incorporate the instructions. We have now fixed all the errors as follows.

i. Speculative sentence on modeling LD in GenoBoost

Reviewer #3 previously encouraged us to investigate why GenoBoost underperformed summary statistics-based methods (Reviewer #3, round 1, point 2). We conducted additional analyses, as suggested, and we are grateful for the suggestion to improve our manuscript.

We now realized that our previous response included seemingly speculative sentences, which we removed in the current version of the manuscript. Specifically, we previously claimed in the discussion section of the manuscript that explicitly incorporating LD information into GenoBoost might improve the predictive accuracy (quoted below).

Although GeneBoost outperformed LDpred, the difference became smaller as the number of causal variants increased, which was consistent with the fact that LDpred performed better than GenoBoost for highly polygenic diseases. Thus, it remains to improve the accuracy of GenoBoost by explicitly modeling the linkage disequilibrium model.

The results of the additional experiments should be better interpreted by the difference in polygenicity and heritability, as suggested by the reviewer, and that the idea that incorporating LD information improves predictive accuracy is speculative. In the revised manuscript, we have removed the sentence from the revision.

In the revised manuscript, we discussed the difference in predictive performance between GenoBoost and LDpred based on the results of simulation study across a wide range of heritability and polygenicity. The results are now shown in **Supplementary Fig 10** and results (quoted below).

Results (page 4, lines 177):

The impacts of non-additive effects at the MHC locus and polygenicity on the best PGS models

To further test the effects of polygenicity on the predictive performance, we generated 80 synthetic phenotypes with varying levels of polygenicity and heritability, applied GenoBoost and LDpred⁵, and compared their predictive performance (Supplementary Fig 10, Supplementary Note 2). Across the polygenicity range tested, GenoBoost consistently outperformed LDpred, the best-performing method without the sparsity constraints. The difference in the predictive performance between GenoBoost and LDpred is larger in less polygenic scenarios.

The conclusion from the suggested additional analysis is now included in the discussion section and the abstract of the manuscript (quoted below).

Discussion (page 6, lines 299):

*We show that GenoBoost improves prediction for autoimmune diseases by incorporating non-additive effects in the MHC locus and, **more broadly, works best in less polygenic traits.***

Abstract (page 1, lines 20):

*We reveal that GenoBoost improves prediction for autoimmune diseases by incorporating non-additive effects localized in the MHC locus and, **more broadly, works best in less polygenic traits.***

ii. Speculative statement on GenoBoost's solving missing heritability

Reviewer #2 pointed out that a paper showed that the linear additive models seem to capture the complete heritability (Reviewer #2, round 1, point 2) and that we should tone down the statement that GenoBoost will be an indispensable approach for settling the missing heritability (Reviewer #2, round 1, point 9). We previously stated that GenoBoost might be useful to investigate the missing heritability (quoted below).

Thus, we believe that GenoBoost is a useful tool to investigate if the missing heritability of a trait⁴⁴ can be attributed to the presence of a non-linear component in the genetic liability and the hypothesis that low-frequency variants explain much of the remaining heritability⁴⁵ when whole-exome or whole-genome sequence data from a statistically sufficient number of samples are available.

However, we now agree that the statement is rather speculative, and we have eliminated the sentence in the previous manuscript.

iii. The difference between the effects estimated by GenoBoost and the association effects

Reviewer #3 (round 1, point 2) pointed out that the effects estimated by GenoBoost are different from association effects. We previously answered that those two effects are almost identical under a limited situation where effects in different LDs were independent. However, the assumption is speculative, and we removed the sentence in the previous manuscript.

Response to Reviewers' comments from previous rounds of review (important note: these points have already been previously addressed)

Our previous responses have already addressed all of the comments below.

However, to avoid any ambiguity as to whether the final form of the manuscript also addresses these comments, we hereby provide an updated point-by-point response, which refers to the figures, supplementary items, and text in the revised latest version of our manuscript.

A large fraction of points have already been addressed and our previous responses are not affected by the current revision, which are shown in gray. Some of them are changed, these are shown in blue.

[Revised response] We show a summary of the updated response in blue.

[Responses not affected by the current revision] we reiterate most of our responses in gray.

Response to Reviewer #1 comments (round 1):

Reviewer #1: In this work, Ohta and colleagues extended the LogitBoost framework for modelling non-additive genetic effects, in order to improve the predictive performance of polygenic risk scores. The authors applied their method, GenoBoost, to predict risks of 12 diseases in the UK Biobank, and demonstrated improved prediction accuracy for some of these diseases achieved by non-additive effect-inclusive models. The manuscript is clearly written. However, I have a number of concerns, mostly regarding the generalizability of the method and some of the interpretations.

We thank the reviewer for the excellent summary of our work and for their helpful and constructive feedback on our manuscript.

1. First of all, GenoBoost achieved the best predictive performance only when predicting five autoimmune/inflammatory diseases (asthma, gout, inflammatory bowel disease, rheumatoid arthritis, and psoriasis), which are correlated with each other. For other diseases that are more polygenic, including cancer (breast cancer and colorectal cancer), cardiovascular (atrial fibrillation and coronary artery disease), and metabolic (type 2 diabetes) diseases, GenoBoost was consistently outperformed by LDPreD, and occasionally PRS-CS. Therefore, I am not convinced that GenoBoost will have broad utility for a wide variety of diseases and am not sure to what extent genetic effect compositions in Figure 3a can be interpreted. Could the authors conduct simulation studies to examine the impacts of (i) disease heritability and (ii) polygenicity (number of causal variants and per-variant effect size) on the performance of GenoBoost?

We have previously revised the manuscript based on your helpful suggestions.

First, we showed that the five inflammatory diseases (asthma, gout, inflammatory bowel disease, rheumatoid arthritis, and psoriasis) have limited phenotypic correlations (pairwise Jaccard similarity index < 0.04) (now in **Supplementary Figure 3** and **Supplementary Notes**).

Second, we conducted a simulation study to investigate the effects of trait heritability and polygenicity, as recommended. The results are now shown in **Supplementary Figure 10**, and we now describe the results of the simulation study in the results section (quoted below).

Results (page 4, lines 172):

The impacts of non-additive effects at the MHC locus and polygenicity on the best PGS models

We also found that the polygenicity of the trait also influences the best-performing PGS models, and our sparse GenoBoost PGS model performs the best for less polygenic traits. [...] To further test the effects of polygenicity on the predictive performance, we generated 80 synthetic phenotypes with varying levels of polygenicity and heritability, applied GenoBoost and LDpred⁵, and compared their predictive performance (Supplementary Fig 10, Supplementary Note 2). Across the polygenicity range tested, GenoBoost consistently outperformed LDpred, the best-performing method without the sparsity constraints. The difference in the predictive performance between GenoBoost and LDpred is larger in less polygenic scenarios.

2. Related to the comment above, as the authors mentioned, MHC region plays an important role in autoimmune/inflammatory diseases. Extended Figures 4 and 5 showed that modelling the MHC region can be challenging for other methods, especially LDpred. Could the authors compare the performance of methods excluding predictors on Chr6, to assess whether the improved prediction accuracy for these diseases was due to better modelling of MHC? The implementation of GenoBoost to avoid extreme SNV values is an important contribution, but may not support the use of this method to predict more polygenic diseases.

We have previously revised the manuscript based on your helpful suggestions.

First, we evaluated the predictive performance of GenoBoost and LDpred without using genetic variants on chromosome 6, as requested. We show the results now in **Supplementary Figure 5** and we now describe the results of the additional analysis in the results section (quoted below).

Results (page 4, lines 158):

The impacts of non-additive effects at the MHC locus and polygenicity on the best PGS models

To evaluate the contribution of non-additive genetic dominance effects at the MHC locus, we prepared additional sets of Additive and Non-additive GenoBoost models for autoimmune disease outcomes without using genetic variants on chromosome 6. Specifically, we removed genetic variants on chromosome 6 from GenoBoost models and additionally constructed another set of GenoBoost models without considering chromosome 6 in model training. In both cases, we found that the predictive performance of Additive and Non-additive GenoBoost models are largely consistent (Supplementary Fig 5, Supplementary Note 1), [...]

The analysis helped us clarify that GenoBoost shows improved predictive performance for autoimmune diseases by incorporating non-additive effects localized in the MHC locus.

The insights we learned from the additional analysis are now described in the abstract and discussion (quoted below).

Abstract (page 1, lines 20):

We reveal that GenoBoost improves prediction for autoimmune diseases by incorporating non-additive effects localized in the MHC locus and, more broadly, works best in less polygenic traits.

Discussion (page 6, lines 293):

*We present GenoBoost, a flexible polygenic modeling framework capable of incorporating both additive and non-additive genetic dominance effects without requiring prior knowledge of the genetic architecture of the trait. Systematic benchmarking of GenoBoost against seven commonly used PGS methods demonstrates the competitive predictive performance of GenoBoost, although no single method outperforms the other approaches across the panel of twelve disease outcomes in UK Biobank evaluated in our study. Taking advantage of the sparsity of GenoBoost models, we investigate the factors that may influence the relative predictive performance of PGS methods. **We show that GenoBoost improves prediction for autoimmune diseases by incorporating non-additive effects in the MHC locus and, more broadly, works best in less polygenic traits.***

3. The authors compared GenoBoost to LASSO using individual-level data. Could a comparison to nonadditive effect-inclusive LASSO models be added? That could simply be $Z \sim G + X$, where G is the genotype (0/1/2) and $X = 1$ if $G = 1$; $X = 0$ if $G = 0$ or 2 , which would only double the degree of freedom. This may demonstrate the advantage of having the boosting framework.

We have previously revised the manuscript based on your helpful suggestions. Specifically, we previously added a variant of LASSO models capable of incorporating non-additive genetic dominance effects, as suggested. We show the results now in **Supplementary Figure 11** and we now briefly describe the results (quoted below).

Results (page 4, lines 205):

Comparison among PGS methods on the individual-level data

To quantify the benefits of non-additive genetic dominance effects in a penalized regression framework, we considered a variant of snpnet capable of considering both additive and non-additive effects and evaluated the predictive performance (Supplementary Methods). We found that non-additive snpnet models improved prediction for rheumatoid arthritis and psoriasis over additive snpnet models (Supplementary Fig 11, Supplementary Note 3). Nonetheless, GenoBoost outperformed non-additive snpnet, highlighting its competitive advantage.

4. The disease outcomes in the UK Biobank mostly have a highly imbalanced case-control ratio. The authors should add AUPRC as an additional metric for evaluating model performance.

We have previously revised the manuscript based on your helpful suggestions. Specifically, we previously added AUPRC as an additional metric of predictive performance. We now show the results in **Supplementary Figure 4c**.

5. The portability of risk scores created by GenoBoost has not been tested in other cohorts or in non-European ancestry sub-populations in the UK Biobank. Would the inclusion of more complicated genetic effects exacerbate the portability?

We have now fixed critical errors in our previous response and in the manuscript, as shown in the “description of the major changes necessary to resolve scientific inaccuracy” section above.

6. The limitations that GenoBoost only handles binary outcomes and relies on availability of individual-level data should be discussed.

We have previously revised the manuscript based on your helpful suggestions.

In the revised text, we clarified that the current implementation of GenoBoost supports binary phenotypes alone and requires individual-level data in the discussion section (quoted below).

Discussion (page 7, lines 331):

*There are several directions for future studies. **First, the current implementation of GenoBoost supports binary phenotypes alone**; future expansion of the methodology into quantitative traits would be helpful. **Second, our current approach operates directly on the individual-level data**; further extension of our method coupled with the increasing availability of non-additive GWAS summary statistics would expand the opportunity for joint modeling of additive and non-additive genetic effects across multiple traits and cohorts.*

Response to Reviewer #1 comments (round 2):

Reviewer #1: The authors have addressed my comments.

Response to Reviewer #2 comments (round 1):

Reviewer #2: Ohta and colleagues propose an innovative method (GenoBoost) for polygenic prediction taking into account both additive and non-additive effects. GenoBoost is based on the boosting framework and the prediction performances were also compared with different alternative PGS approaches showing competitive results and identifying potential candidate variants with non-linear (e.g., dominant and recessive) effect. The manuscript is well written and the analysis can be of high interest in the genetic epidemiology community. However, the reviewer believe that a few major points should be considered to improve the work.

We thank the reviewer for the excellent summary of our work and for their helpful and constructive feedback on our manuscript.

In particular:

1. Despite comparisons with different PRS methods based on summary statistics are reported, there is only one comparison with a tool based on multi-variable regression directly from genotyping data, namely snpnet which exploits the lasso framework. Since starting from genotyping provide more information with respect to GWAS, I would suggest to include another individual-level genotype PGS model approach, e.g., snpboost tool (<https://github.com/hklinkhammer/snpboost>) which could represent an optimal comparison by being also based on boosting framework to derive sparse additive PGS models.

We have previously revised the manuscript based on your helpful suggestions. Specifically, we previously added snpboost to our comparison of PGS models. We now show the results in **Figure 2** and **Supplementary Figure 4**.

Moreover, we now present a subsection in results, titled “Comparison among PGS methods on the individual-level data,” to highlight the differences across three PGS methods on the individual-level data (quoted below).

Results (page 4, lines 187):

Comparison among PGS methods on the individual-level data

We next compared GenoBoost against snpnet^{20} and snpboost^{27} , the two recently developed PGS methods that directly operate on individual-level data. The comparison demonstrated the unique advantage of GenoBoost in incorporating non-additive genetic dominance effects and also resulting in extremely sparse PGS models. Across twelve traits, the three PGS methods on the individual-level data have fewer genetic variants than those based on summary statistics (Fig 2d, Supplementary Figs 4e, 8). Indeed, GenoBoost selected the least number of genetic variants for ten out of twelve traits. Among the three methods on the individual-level data, GenoBoost has the highest level of sparsity with the median of 195 genetic variants, while snpnet and GenoBoost selected the median of 913 and 727 genetic variants across twelve traits.

We compared GenoBoost against snpboost^{27} , given that both methods are built on statistical boosting. Overall, GenoBoost showed improved predictive performance over snpboost across eight out of twelve traits, in which our methodological advancements in statistical boosting likely played a substantial role. Specifically, our theoretical results on the analytical solution for optimal GenoBoost score enabled computationally efficient PGS modeling while allowing us to consider both additive and non-additive genetic dominance effects.

We showed 4.9% average improvements across the twelve traits over snpnet^{20} , an implementation of batch screening iterative lasso on large-scale genetic datasets while considering additive effects alone (Fig 2a). For Alzheimer's disease and all-cause dementia, snpnet showed the best predictive performance (covariate-adjusted pseudo- $R^2=0.0408$ and 0.0344 , respectively), followed by GenoBoost with a small difference in predictive performance (pseudo- $R^2=0.0408$ and 0.0342 , respectively), highlighting the competitive performance of GenoBoost. To quantify the benefits of non-additive genetic dominance effects in a penalized regression framework, we considered a variant of snpnet capable of considering both additive and non-additive effects and evaluated the predictive performance (Supplementary Methods). We found that non-additive snpnet models improved prediction for rheumatoid arthritis and psoriasis over additive snpnet models (Supplementary Fig 11, Supplementary Note 3). Nonetheless, GenoBoost outperformed non-additive snpnet , highlighting its competitive advantage.

We also discuss the methodological differences between GenoBoost (our proposal) and snpboost in the discussion section (quoted below).

Discussion (page 6, lines 307):

Methodologically, our study provides substantial advancements in PGS modeling efforts for complex traits. For example, to the best of our knowledge, GenoBoost is the first PGS method that can consider the effects of additive and non-additive genetic dominance effects simultaneously. The advantage of our analytical solution is evident when we compare GenoBoost against snpboost²⁷, a recently developed PGS method based on the application of statistical boosting on the individual-level genetic data: snpboost is based on L₂-boosting⁵⁶, a specific kind of statistical boosting technique best suited for quantitative traits, and is limited to additive effects; GenoBoost, in contrast, builds on LogitBoost, which was originally developed for binary classification³³, and also on our theoretical results. The analytical solutions for the optimal GenoBoost scores (Supplementary Methods) overcome the limitations in previous approaches and allow efficient inference of additive and non-additive genetic dominance effects.

2. Concerning the missing heritability issue (gap between observed heritability and the one explained by additive SNP-effect) there is an open discussion regarding the potential sources: including nonlinear effect, rare-variants role and gene-environment interactions. Recent works suggest that at least for highly polygenic trait (e.g., height) once there is enough data the linear additive models seem to capture the complete heritability due to common variants (<https://www.nature.com/articles/s41586022-05275-y>). In this regard the reported results in which non-linear GenoBoost significantly outperforms linear GenoBoost (i.e., rheumatoid arthritis and psoriasis) would be of high interest. Could authors check if there is an (inverse) correlation between the polygenicity of the signal (e.g., number of selected variants) and the proportion of non-linear detected variants. This would align with the current literature suggesting that highly polygenic traits are mainly characterized by additive effects at the level of common variants whereas low polygenic traits can be characterized by a sparse genetic architecture including also not-negligible non-linear effects.

We have now fixed critical errors in our previous response and in the manuscript, as shown in the “description of the major changes necessary to resolve scientific inaccuracy” section above. Briefly,

- We removed our seemingly speculative remarks on “missing heritability issue” in the manuscript;
- We clarified that there was a modest correlation between the number of genetic variants included in the GenoBoost models (approximate measure of polygenicity) and the proportion of genetic variants with inferred non-additive genetic dominance effects; and
- We now clarified that GenoBoost works best in less polygenic traits.

3. For the validation of the detected variants with non-linear effects authors considered the closest genes and evaluated gene-phenotype associations from literatures. This is definitely a valid approach to corroborate the findings, however, I would suggest given the availability of genotype/phenotype data to test dominant and recessive models via univariable association directly on the genotyping data. This would allow a direct comparison of the selected PGS variant, in fact, the variant-gene match especially in non-coding regions might be quite complex in terms of biological interpretation (e.g., one could consider also eQTL and other regulatory mechanisms).

We have previously revised the manuscript based on your helpful suggestions.

First, for all of the genetic variants selected in our GenoBoost model, we conducted univariate association tests under non-additive inheritance models, as suggested. We showed, in **Figure 3b**, that the inferred mode of inheritance from two orthogonal approaches (GenoBoost and univariate tests) are largely consistent.

In the current manuscript, we improved the clarity of the methodological description of the univariate tests as well as the interpretation of the results (quoted below).

Results (page 5, lines 243):

GenoBoost scores allow inference of the mode of inheritance

We validated that the inferred mode of genetic inheritance by GenoBoost is highly consistent with what one may infer from the GWAS association statistics conducted under the various genetic inheritance modes^{12,14}. Specifically, we conducted GWAS analysis for each genetic variant under additive and all non-additive genetic dominance modes and inferred the genetic inheritance mode to be the one that resulted in the most significant association summary statistics. We found the inferred inheritance modes from the two approaches are largely consistent, although the two approaches rely on different metrics (Fig 3b).

Methods (page 8, lines 409):

Genome-wide association analysis

[...] To infer the mode of genetic inheritance using the statistical significance of GWAS associations, we repeated the GWAS analysis with PLINK using the following list of modifiers in the "--glm" command: "dominant", "recessive", and "hetonly".

Second, we highlighted examples of non-additive genetic dominance effects in non-coding regions, following your suggestion. Specifically, we previously highlighted *MICD* pseudogene as an example. The genomic region was reported in the literature (He, P. et al. 2021, PubMed ID: 35386192, reference [55] in our manuscript) and potentially has pleiotropic effects for immune and skeletal disease including psoriasis.

We now show the results in **Supplementary Figures 15-16** and describe the example in the results section (quoted below).

Results (page 6, lines 283):

Applying GenoBoost to prioritize genetic loci previously not reported in the GWAS catalog

*[...] Beyond the gene body, we found a genetic variant in the *MICD* pseudogene, which was reported to be a potential pleiotropic gene for immune and skeletal disease, including psoriasis⁵⁵ (Supplementary Data 1).*

4. There are a few traits in which GenoBoost performances are clearly lower than the best alternative performing method (e.g., coronary artery disease and type 2 diabetes). Authors claims this is due to the fact that the alternative Bayesian approaches by incorporating the LD can better compute the heritability of the

regions. Would it be possible to analyze and report how the model sparsity scale with respect to the prediction performance (e.g., by evaluating the different learning rate grid results) to check that indeed these are the cases in which the prediction performances have not yet reached a maximum (suggesting that such traits are highly polygenic and thus sparse models are not optimal in terms of prediction in comparison to “omnigenic” models).

We have previously revised the manuscript based on your helpful suggestions. Specifically, we confirmed that the prediction performance reached the maximum. The results are now shown in **Supplementary Figure 18**, and we now describe the results in the methods section (quoted below).

Applying GenoBoost to UK Biobank disease outcomes

[...] The maximum number of unique selected genetic variants is constrained to be 10,000, given the maximum iteration count, T , tested in the grid search. We confirmed that the cross-validation selected less than 10,000 genetic variants (Supplementary Fig 18).

Moreover, we now eliminated the speculative statement that explicitly incorporating LD information into GenoBoost would improve predictive accuracy, as shown in the “description of the major changes necessary to resolve scientific inaccuracy” section above.

Minor comments:

5. I would suggest to rephrase the 1st sentence in the abstract: I would skip the word “improve” as there is no comparison but rather just indicating that “PGS can be used for phenotype prediction by aggregating...” similarly, I would avoid the term “however” as there is no contrast but rather directly report the nonlinear effects are typically neglected in PGS models.
6. The acronym SNV was not initially defined
7. I would remodulate a little some sentences referring to the relevance of non-linear effects and GenoBoost: e.g. “Our results demonstrate that the non-additive model is essential in predicting risk for polygenic diseases.”
8. For some of the reported phenotypes that is not what can be observed in the results, I would rather write that according to genetic architecture of the underlying traits non-linear models can improve prediction and understanding of the genetic liability.
9. “we believe that GenoBoost will be an indispensable approach for settling the “missing heritability” problem.” I would instead indicate that GenoBoost can be a useful tool to investigate if the missing heritability of a traits can be attributed the presence of a non-linear component in the genetic liability

We have previously fixed all of the issues and we now confirm that our revised manuscript does not have any of the issues.

Response to Reviewer #2 comments (round 2):

Reviewer #2: I thank the authors for thoroughly addressing all of my concerns. I believe that the additional analysis and interpretation have significantly improved the manuscript's quality. I believe this work represents a valuable contribution to the field of statistical genetics in the context of polygenic risk modeling. I have no further issues to raise.

Response to Reviewer #3 comments (round 1):

Reviewer #3: The authors proposed a novel PRS method that integrates both additive and non-additive genetic effects using the boosting algorithm. In the UKB data application, for certain traits that exhibit nonadditive genetic architecture, the method showed improved performance as compared to existing PRS approaches.

The idea to incorporate non-additive effects to PRS is novel and optimization through boosting, which has been trending in PRS method research (PMIDs: 28979001, 36704342, 35995843), is a clever approach to combine multiple weak classifiers as often is the case for GWAS signals. However, I find the conclusion that “the non-additive model is essential in predicting risk for polygenic diseases” not founded as non-additive genetic variance contribute very little to complex traits at large as demonstrated by many others (Palmer et al., 2023 - 36996212, Hivert et al., 2021 - 33811805). I do believe that non-additive model is important to study but have doubts about whether the proposed PRS can achieve what the authors suggested, in replacing other additive PRSs that have better performance on complex polygenic traits that have sizeable genetic effects. The authors should focus on defining for which traits their method is expected to excel; the results are already there, but the message is not clear yet. For example, contrast the good results (RA, Asthma, etc) with the bad (T2D, CAD, etc) and really hone in on what makes their method work and otherwise.

We have previously revised the manuscript based on your helpful suggestions.

First, we previously removed our seemingly speculative sentences like “the non-additive model is essential in predicting risk for polygenic diseases,” as suggested.

Second, in the current revised manuscript, we substantially improved the clarity of the text, aiming to (1) contextualize our manuscript in light of the literature reporting the limited roles of genetic dominance heritability, as suggested and (2) clarify the situation where the proposed GenoBoost model is the most useful, as requested.

(1) Improved description of earlier studies on boosting and genetic dominance heritability

In the introduction, our revised manuscript cites the suggested papers regarding statistical boosting method and dominance heritability.

Introduction (page 2, lines 57):

To account for genetic dominance not captured in additive GWAS summary statistics, GenoBoost directly operates on individual-level data, building on top of the recent methodological innovations by us and others on PGS modeling, including penalized regression¹⁷⁻²⁵ and statistical boosting²⁶⁻³⁰, both relying on individual-level data.

Introduction (page 1, lines 41):

Earlier studies focused on a smaller number of samples and estimated that the relative contribution of non-additive effects on trait heritability is smaller than that of additive effects⁹. However, the increase in sample sizes in recent studies coupled with methodological advancements started to demonstrate the presence of non-additive heritability, and their relative contribution varies depending on the genetic architecture of traits¹⁰⁻¹⁴. For example, some highlight substantial roles of non-additive effects in autoimmune diseases, most notably in rheumatoid arthritis and psoriasis¹⁵, suggesting the potential benefits of considering non-additive effects in polygenic score modeling for some traits.

(2) improved description of when GenoBoost is the most useful

We clarified the characteristics of phenotypes where GenoBoost is competitive with other PGS methods or had improved predictive accuracy in abstract, results, and discussion.

Abstract:

We reveal that GenoBoost improves prediction for autoimmune diseases by incorporating non-additive effects localized in the MHC locus and, more broadly, works best in less polygenic traits.

Results (page 3, lines 147):

The impacts of non-additive effects at the MHC locus and polygenicity on the best PGS models

We focused on three immune-related disorders, where GenoBoost showed the best predictive performance with the most improvements over the second-best methods. Specifically, our Non-additive GenoBoost performed the best for the two autoimmune diseases, rheumatoid arthritis and psoriasis, with covariate-adjusted pseudo- R^2 of 0.0129 and 0.0436, showing 11.3% and 2.78% improvements over snpboost²⁷, the second-best performing PGS methods (Fig 2b, Supplementary Fig 4a). Additive GenoBoost was ranked the best for gout (pseudo- $R^2=0.0401$), showing 2.09% improvements over snpboost.

We found that the improvements in the predictive performance for autoimmune disorders with GenoBoost can be attributed to non-additive genetic dominance effects in a relatively small number of genetic loci. For example, genetic variants in the major histocompatibility complex (MHC) locus in chromosome 6 contribute substantially to predicting the genetic liability of autoimmune diseases in the GenoBoost model. The heterozygosity of the MHC regions is protective against psoriatic arthritis⁴⁷, and some reports indicate over-dominant selection at the locus⁴⁸. To evaluate the contribution of non-additive genetic dominance effects at the MHC locus, we prepared additional sets of Additive and Non-additive GenoBoost models for autoimmune disease outcomes without using genetic variants on chromosome 6. Specifically, we removed genetic variants on chromosome 6 from GenoBoost models and additionally constructed another set of GenoBoost models without considering chromosome 6 in model training. In both cases, we found that the predictive performance of Additive and Non-additive GenoBoost models are largely consistent (Supplementary Fig 5, Supplementary Note 1), highlighting the unique ability of GenoBoost to incorporate non-additive genetic dominance effects at the MHC locus in improving genetic risk prediction of autoimmune disorders.

We hypothesized that the moderate regularization of non-additive genetic dominance effects in GenoBoost would further improve the predictive performance, given the lower frequency of homozygous carriers in populations and limited roles of non-additive effects beyond the MHC locus. We introduced the maximum absolute value for the GenoBoost s_2 score, representing the effect of homozygous minor alleles in the GenoBoost model. We indeed found that the modest regularization improved the predictive performance (Supplementary Figs 6-7).

We also found that the polygenicity of the trait also influences the best-performing PGS models, and our sparse GenoBoost PGS model performs the best for less polygenic traits. We first investigated the results for psoriasis and asthma, where GenoBoost ranked second (pseudo- $R^2=0.0308$) after snpboost²⁷ (pseudo- $R^2=0.0310$), given the known role of the MHC locus in both traits and the difference between the two traits in the polygenicity (Supplementary Fig 8). The distribution of the PGS scores in the held-out test set individuals also reflects the difference in polygenicity (Supplementary Fig 9). To further test the effects of polygenicity on the predictive performance, we generated 80 synthetic phenotypes with varying levels of polygenicity and heritability, applied GenoBoost and LDpred⁵, and compared their predictive performance (Supplementary Fig 10, Supplementary Note 2). Across the polygenicity range tested, GenoBoost consistently outperformed LDpred, the best-performing method without the sparsity constraints. The difference in the predictive performance between GenoBoost and LDpred is larger in less polygenic scenarios. In our benchmarking on the twelve UK Biobank disease outcomes, LDpred was ranked as the best-performing PGS method for highly polygenic traits, such as atrial fibrillation, breast cancer, and coronary artery disease, suggesting that sparse PGS methods like ours are most advantageous for less polygenic traits.

Discussion (page 6, lines 299):

We show that GenoBoost improves prediction for autoimmune diseases by incorporating non-additive effects in the MHC locus and, more broadly, works best in less polygenic traits.

Specific comments:

1. In the abstract, I find it misleading to say that 40-67% of variants were non-additive in the GenoBoost model since only a small # of SNVs were present in the model to start with. This is obviously one limitation of the method, setting the maximum number of SNVs to be 10,000. Have the authors investigated whether increasing this # would improve performance for traits where it currently underperforms? More specifically, how much of the small # of SNVs included in the final model was driven by Boosting and how much by the underlying genetic architecture of the trait?

We have previously revised the manuscript based on your helpful suggestions.

First, we previously removed our seemingly speculative phrases like “approximately 40-67% of variants, including some variants on disease-associated genes, were non-additive in the GenoBoost models” in the abstract, as suggested.

Second, we previously confirmed that the prediction performance already reached the maximum, and there was no further increase in predictive performance when we allowed the model to consider a larger number of genetic variants. We now show the results in **Supplementary Figure 18**.

2. If GenoBoost also considers additive models, then why does it still underperform additive models like lassosum and LDpred? On the other hand, why is the additive GenoBoost better than the other additive models for certain traits?

We now eliminated the speculative statement that incorporating LD to GenoBoost would improve predictive accuracy, as shown in the “description of the major changes necessary to resolve scientific inaccuracy” section above.

3. As for its application to characterize genetic model, the idea to use PRS to explore whether nonadditive variance vs. additive variance is the primary contributor is interesting. But I have a hard time agreeing that the effects estimated from GenoBoost can be considered association effects (line 173-174). These optimized effects were for prediction and should be followed up by a formal testing if any claims were to be made. In any case, the strength of the method is through the aggregated nonadditive genetic effects, testing individual SNPs requires sample size much larger as pointed out by Palmer et al. (2023).

We thank the reviewer for pointing out inappropriate expressions. We removed the sentence mentioning the relationship between GenoBoost’s effects and association effects as shown in the “description of the major changes necessary to resolve scientific inaccuracy” section above. We confirmed that no claims were made based on the sentence in the current version of the manuscript.

4. It is not clear how the authors selected these 12 traits and I don’t think these are “representative”, at least in comparison to what were typically used in the large collection of PRS methods papers. - The authors need to tone down the conclusion and be careful of the wording. For example, GenoBoost cannot “classify” genetic model (line 203), it simply selects the model that improves prediction. And the authors have not shown it to be “essential” in predicting polygenic disease in which only 5 of the 12 were best.

We thank the reviewer for pointing out inappropriate expressions. In the revised manuscript, we addressed all of the concerns.

(1) Improved description of twelve disease outcomes in UK Biobank analyzed in the study

We agree with the reviewer that the word “representative” is inappropriate. We removed the word “representative” from the sentence. We clarified how to select 12 traits.

Results (page 3, lines 117):

Benchmarking GenoBoost across twelve disease outcomes in UK Biobank

We applied GenoBoost and seven previously published PGS methods to the twelve disease outcomes and evaluated their predictive performance (Table 1). In the panel of twelve disease outcomes in UK Biobank^{35,36}, we included seven commonly studied traits (rheumatoid arthritis, inflammatory bowel disease, asthma, atrial fibrillation, breast cancer, coronary artery disease, and type 2 diabetes) in the PGS literature^{5,7,8,20,39–41} and five disease outcomes (psoriasis⁴², gout⁴³, all-cause dementia⁴⁴, Alzheimer’s disease⁴⁵, and colorectal cancer⁴⁶) with high prevalence and genetic basis (Methods)³⁶. Among the twelve traits selected, rheumatoid arthritis and psoriasis are included in the list of candidate phenotypes with substantial non-additive genetic dominance effects in a recent study¹⁵.

(2) Inferring the mode of inheritance of genetic variants with GenoBoost

We agree with the reviewer that the original expression — “Non-additive GenoBoost has the ability to classify genetic inheritance models and prioritize the non-additive variants” — was inappropriate. In the revised manuscript we now clarify that Non-additive GenoBoost infers the mode of genetic inheritance for each genetic variant from three GenoBoost scores (s_0 , s_1 , and s_2). The revised text in the results section in the manuscript is quoted below.

Results (page 5, lines 232):

GenoBoost scores allow inference of the mode of inheritance

The three GenoBoost scores for each genetic variant (s_0 , s_1 , and s_2) in the sparse GenoBoost PGS models allow us to classify the genetic variant effects based on the types of the estimated inheritance mode into additive, dominant, recessive, over-dominant, and over-recessive (Supplementary Fig 2, Supplementary Methods).

(3) Inappropriate claim that non-additive effects are “essential” in polygenic risk modeling

We agree with the reviewer that the word “essential” is inappropriate. In the revised manuscript, we now removed the word “essential” from the abstract and discussion.

Abstract (page 1, lines 20):

We reveal that GenoBoost improves prediction for autoimmune diseases by incorporating non-additive effects localized in the MHC locus and, more broadly, works best in less polygenic traits.

Discussion (page 7, lines 342):

Our results highlight the benefits of incorporating non-additive genetic dominance effects in PGS models. We demonstrate the advantage of our sparse PGS models in improving predictive performance for autoimmune disorders and less polygenic traits with non-additive effects.

5. Extreme SNV scores: this winsorizing of the effect was to deal with the current sample size, can the authors comment on the choice of the adjustment and how different choices of adjustment value impact the fit/final results?

We have previously revised the manuscript based on your helpful suggestions.

First, we clarified how to winsorize the effect size estimates in the GenoBoost model. The descriptions are now found in **Supplementary Figure 6** and **Supplementary Methods** (quoted below).

Supplementary Methods (page 11):

1.7 Limiting extremely large absolute SNV scores

The threshold $4 ds_{t,1}$ is the one dividing $ds_{t,1}$ into additive mode and dominant mode at equal intervals, and other thresholds are set so that estimated additive, dominant, and recessive modes have equal range of $ds_{t,1}$ (Supplementary Fig. 2).

Second, we previously confirmed that current adjustment had higher predictive accuracy than another adjustment (**Supplementary Table 2**) and no adjustment and we keep the results in the current version of the manuscript in **Supplementary Figure 7**.

We now clarify those two points in the main text (quoted below).

Results (page 4, lines 166):

The impacts of non-additive effects at the MHC locus and polygenicity on the best PGS models

We hypothesized that the moderate regularization of non-additive genetic dominance effects in GenoBoost would further improve the predictive performance, given the lower frequency of homozygous carriers in populations and limited roles of non-additive effects beyond the MHC locus. We introduced the maximum absolute value for the GenoBoost s_2 score, representing the effect of homozygous minor alleles in the GenoBoost model. We indeed found that the modest regularization improved the predictive performance (Supplementary Figs 6-7).

6. Finally, correction modelling of LD or models that are robust to LD misspecification have been shown to be the key in improving PRSs, thus, I find the discussion on why GenoBoost is unable or how to incorporate in future research too brief. For example the authors might want to address why in some cases (e.g. RA), ignoring LD seems to be okay but in others this has the consequence of worse performance as compared to other methods.

We thank the reviewer for pointing out inappropriate expressions and encouraging us to discuss further on the future research directions. We addressed them in the revised manuscript.

First, we now eliminated the speculative statement that explicitly incorporating LD information into GenoBoost would improve predictive accuracy, as shown in the “description of the major changes necessary to resolve scientific inaccuracy” section above. We, instead, provided the interpretation of the results based on the differences in heritability and polygenicity of the traits.

Second, we expanded biological interpretation on when GenoBoost is the most effective. Specifically, our revised manuscript now includes the results subsections focusing on targeted comparison of different PGS methods. Those subsections include: “***The impacts of non-additive effects at the MHC locus and polygenicity on the best PGS models***” and “***Comparison among PGS methods on the individual-level data***.” The insights from the comparisons are now clearly stated in the manuscript, most notably in the abstract (quoted below).

Abstract (page 1, lines 20):

We reveal that GenoBoost improves prediction for autoimmune diseases by incorporating non-additive effects localized in the MHC locus and, more broadly, works best in less polygenic traits.

Third, we now expanded the future research in the discussion (quoted below).

Discussion (page 7, lines 331):

There are several directions for future studies. First, the current implementation of GenoBoost supports binary phenotypes alone; future expansion of the methodology into quantitative traits would be helpful. Second, our current approach operates directly on the individual-level data; further extension of our method coupled with the increasing availability of non-additive GWAS summary statistics would expand the opportunity for joint modeling of additive and non-additive genetic effects across multiple traits and cohorts. Third, our study focused on common SNVs alone; future studies should incorporate a broader type of genetic variants, including indels, HLA allelotypes⁵⁷, copy number variations, short tandem repeat expansions, microsatellites, and structural variants, given the increasing catalog of rare genetic variants from short- and long-read sequencing-based studies and methodological innovations of integrating those effects in genetic scores⁵⁸. Fourth, our current model did not consider the interaction between genetic variants; future non-additive PGS models should incorporate such effects.

Minor comments:

7. Acronyms spell out the first time when mentioning, e.g. GWAS, SNV, etc.

We have previously fixed issues regarding the acronyms and confirmed that our revised manuscript does not have the same issue.

8. sBayes is missing for a few traits (Psoriasis, AD) in Figure 1.

We clarified that SBayesR raised convergence error for some phenotypes in the legend of **Figure 2**.

SBayesR raised convergence error for psoriasis and Alzheimer's disease and for four out of five cross-validations of gout.

9. The light pink color in Figure 3-C and D was hard to see.

We have previously fixed the color issue in the previous manuscript.

10. Figure 7-8 really difficult to read, also maybe consider collapsing; these p-values are really not that different - and neither passed threshold for significant testing, maybe shown in a table instead?

We previously added **Supplementary Table 1**.

Response to Reviewer #3 comments (round 2):

Reviewer #3: I am overall satisfied with the responses. But there are some remaining gaps that the authors can close to help readers. To contextualize the results, the authors should perhaps stress throughout what is so characteristic about the 12 traits for which genoBoost is competitive with respect to other PRS methods. For starters, it should be clear in the abstract that these are 12 binary phenotypes. Is there any criteria for the selection process that can offer insights? Or specifically for which disease categories that genoBoost seems to offer improved performance, in terms of genetic architecture from existing GWASs.

We thank the reviewer for suggesting improving the clarity of the main text. We clarified the characteristics of phenotypes where GenoBoost is competitive with other PGS methods or had improved predictive accuracy, the selection criteria for 12 traits in the abstract, results, and discussion.

Abstract (page 1, lines 20):

We reveal that GenoBoost improves prediction for autoimmune diseases by incorporating non-additive effects localized in the MHC locus and, more broadly, works best in less polygenic traits.

Results (page 2, lines 100):

Overview of the GenoBoost study

We assembled a panel of twelve disease outcomes with known heritable basis and high prevalence in UK Biobank

Results (page 3, lines 117):

Benchmarking GenoBoost across twelve disease outcomes in UK Biobank

We applied GenoBoost and seven previously published PGS methods to the twelve disease outcomes and evaluated their predictive performance (Table 1). In the panel of twelve disease outcomes in UK Biobank^{35,36}, we included seven commonly studied traits (rheumatoid arthritis, inflammatory bowel disease, asthma, atrial fibrillation, breast cancer, coronary artery disease, and type 2 diabetes) in the PGS literature^{5,7,8,20,39-41} and five disease outcomes (psoriasis⁴², gout⁴³, all-cause dementia⁴⁴, Alzheimer's disease⁴⁵, and colorectal cancer⁴⁶) with high prevalence and genetic basis (Methods)³⁶. Among the twelve traits selected, rheumatoid arthritis and psoriasis are included in the list of candidate phenotypes with substantial non-additive genetic dominance effects in a recent study¹⁵.

The impacts of non-additive effects at the MHC locus and polygenicity on the best PGS models

We focused on three immune-related disorders, where GenoBoost showed the best predictive performance with the most improvements over the second-best methods. Specifically, our Non-additive GenoBoost performed the best for the two autoimmune diseases, rheumatoid arthritis and psoriasis, with covariate-adjusted pseudo- R^2 of 0.0129 and 0.0436, showing 11.3% and 2.78% improvements over snpboost²⁷, the second-best performing PGS methods (Fig 2b, Supplementary Fig 4a). Additive GenoBoost was ranked the best for gout (pseudo- $R^2=0.0401$), showing 2.09% improvements over snpboost.

We found that the improvements in the predictive performance for autoimmune disorders with GenoBoost can be attributed to non-additive genetic dominance effects in a relatively small number of genetic loci. For example, genetic variants in the major histocompatibility complex (MHC) locus in chromosome 6 contribute substantially to predicting the genetic liability of autoimmune diseases in the GenoBoost model. The heterozygosity of the MHC regions is protective against psoriatic arthritis⁴⁷, and some reports indicate over-dominant selection at the locus⁴⁸. To evaluate the contribution of non-additive genetic dominance effects at the MHC locus, we prepared additional sets of Additive and Non-additive GenoBoost models for autoimmune disease outcomes without using genetic variants on chromosome 6. Specifically, we removed genetic variants on chromosome 6 from GenoBoost models and additionally constructed another set of GenoBoost models without considering chromosome 6 in model training. In both cases, we found that the predictive performance of Additive and Non-additive GenoBoost models are largely consistent (Supplementary Fig 5, Supplementary Note 1), highlighting the unique ability of GenoBoost to incorporate non-additive genetic dominance effects at the MHC locus in improving genetic risk prediction of autoimmune disorders.

We hypothesized that the moderate regularization of non-additive genetic dominance effects in GenoBoost would further improve the predictive performance, given the lower frequency of homozygous carriers in populations and limited roles of non-additive effects beyond the MHC locus. We introduced the maximum absolute value for the GenoBoost s_2 score, representing the effect of homozygous minor alleles in the GenoBoost model. We indeed found that the modest regularization improved the predictive performance (Supplementary Figs 6-7).

We also found that the polygenicity of the trait also influences the best-performing PGS models, and our sparse GenoBoost PGS model performs the best for less polygenic traits. We first investigated the results for psoriasis and asthma, where GenoBoost ranked second (pseudo- $R^2=0.0308$) after snpboost²⁷ (pseudo- $R^2=0.0310$), given the known role of the MHC locus in both traits and the difference between the two traits in the polygenicity (Supplementary Fig 8). The distribution of the PGS scores in the held-out test set individuals also reflects the difference in polygenicity (Supplementary Fig 9). To further test the effects of polygenicity on the predictive performance, we generated 80 synthetic phenotypes with varying levels of polygenicity and heritability, applied GenoBoost and LDpred⁵, and compared their predictive performance (Supplementary Fig 10, Supplementary Note 2). Across the polygenicity range tested, GenoBoost consistently outperformed LDpred, the best-performing method without the sparsity constraints. The difference in the predictive performance between GenoBoost and LDpred is larger in less polygenic scenarios. In our benchmarking on the twelve UK Biobank disease outcomes, LDpred was ranked as the best-performing PGS method for highly polygenic traits, such as atrial fibrillation, breast cancer, and coronary artery disease, suggesting that sparse PGS methods like ours are most advantageous for less polygenic traits.

Discussion (page 6, lines 299):

We show that GenoBoost improves prediction for autoimmune diseases by incorporating non-additive effects in the MHC locus and, more broadly, works best in less polygenic traits.

Reviewer #1 (Remarks to the Author):

I thank the authors for correcting the mistakes. The results and interpretations now make more sense. I have no further comments.

Reviewer #1 (Remarks on code availability):

The analyses are sufficiently documented.

Reviewer #2 (Remarks to the Author):

I thank the authors for the relevant and appropriate changes regarding the analysis of PRS with respect to population stratification.

I also thank the authors for the updated computation of the correlation between polygenicity and non-linear effect as well as for revising rather speculative sentences of the manuscript.

I believe this manuscript will be highly valuable to researchers focused on statistical genetics and the genetic prediction of complex, multifactorial polygenic traits.

Reviewer #3 (Remarks to the Author):

The authors did a great job contextualizing their methods among existing alternatives mostly designed for highly polygenic traits under additive influence.

Specific comments:

- 1: since the authors had done the hard work of testing the portability, I think there are a few conclusions that can be drawn that will benefit the community: 1) it seems that for some traits, that the non-additive effect models "exacerbate the portability issue" as commented by R1; such as T2D in SA, Psoriasis in Africans; 2) there are also instances that GenoBoost seems to perform better than other PGS methods; such as Psoriasis in SA and Gout in East Asian. This can be further discussed and may support the use of GenoBoost in specific populations over other PGS methods.

- 2: polygenicity of the signal and nonlinear variants. I was wondering if we should assess the polygenicity of the signal based on variants selected by other PGS methods, for example lassosum or LDpred? I feel GenoBoost selects fewer SNVs overall given the constraint and some readers might incorrectly conclude from this that the more polygenic a trait is, the higher the proportion of SNVs with non-additive effects. This can be done in a few traits where the estimated h^2 is similar between GenoBoost and lassosum/LDpred. A plot of the number of SNVs included in each model would also be useful.

- 3-i: I am a bit troubled by the results shown in Supplementary Figure 10, between a) and b) under the no dominance heritability scenarios. In a), when the overall h^2 is 0.1 and 100 causal variants, LDpred seems comparable to GenoBoost, however, in b) where the only difference is a higher h^2 ($=0.25$), then LDpred only captures 50% of the heritability ($\sim 8\%$) as compared to GenoBoost (h^2 estimate $\sim 16\%$). I wonder if there is some explanation for this. In line with this observation, I think the authors should be specific about that the simulation study did not incorporate LD, a scenario that puts LDpred at a disadvantage.

Minor comments:

- Figure labels and legends incorrect for Supplementary Figure 10: $h^2=0.1$ in a) and $h^2=0.25$ in b)

- Legend of Supplementary Figure 12 '(n=2,975; African, n=2,867; South Asian, n=904)': it is not clear which sample size for which population. I suspect East Asian is for the last one?
- Given these minor inconsistencies, I suggest the authors have a final check on all figure legends and tables.

Reviewer #3 (Remarks on code availability):

The package runs well and has a nice introduction to get hands on the software quickly. I was able to install, run the codes, and train a model using the data supplied.

Author response to editorial and reviewer comments for manuscript NCOMMS-23-28715D titled “**A Polygenic Score Method Boosted by Non-additive Models**” by Ohta, *et al.*

We are grateful for another round of review for the manuscript to resolve scientific inaccuracies. We would like to thank the reviewers for giving us valuable comments and suggestions. Please find answers to all comments below.

Response to Reviewer #1 comments:

I thank the authors for correcting the mistakes. The results and interpretations now make more sense. I have no further comments.

We thank the reviewer for the kind words and the appreciation of our revised manuscript.

Response to Reviewer #2 comments:

I thank the authors for the relevant and appropriate changes regarding the analysis of PRS with respect to population stratification.

I also thank the authors for the updated computation of the correlation between polygenicity and non-linear effect as well as for revising rather speculative sentences of the manuscript.

I believe this manuscript will be highly valuable to researchers focused on statistical genetics and the genetic prediction of complex, multifactorial polygenic traits.

We thank the reviewer for the kind words and the appreciation of our revised manuscript.

Response to Reviewer #3 comments:

The authors did a great job contextualizing their methods among existing alternatives mostly designed for highly polygenic traits under additive influence.

Specific comments:

1: since the authors had done the hard work of testing the portability, I think there are a few conclusions that can be drawn that will benefit the community: 1) it seems that for some traits, that the non-additive effect models “exacerbate the portability issue” as commented by R1; such as T2D in SA, Psoriasis in Africans; 2) there are also instances that GenoBoost seems to perform better than other PGS methods; such as Psoriasis in SA and Gout in East Asian. This can be further discussed and may support the use of GenoBoost in specific populations over other PGS methods.

We thank the reviewer for the suggestion to include the discussion on trans-ancestry portability of PGS models. In the revised manuscript, we now clarify that there is substantial variability in PGS portability and highlight specific example (trait, ancestry) pairs, as suggested.

Results (page 5, lines 230):

*We observed substantial variability in predictive performance in non-European populations across disease outcomes and PGS methods. For example, GenoBoost showed the best predictive performance for psoriasis for the East Asian population and gout for the South Asian population, whereas other methods showed better performance for gout for the African population and type 2 diabetes for the South Asian population, motivating further expansion of GenoBoost into multi-ancestry settings in future studies (**Discussion**).*

We also discuss an extension of GenoBoost to multi-ancestry settings across cohorts as one of the promising directions for future follow-up studies.

Discussion (page 7, lines 347):

Fourth, we currently focus on European individuals alone in UK Biobank as proof of principle, but future studies will benefit greatly by incorporating individuals across diverse genetic ancestry from multiple cohorts^{25,59}.

2: polygenicity of the signal and nonlinear variants. I was wondering if we should assess the polygenicity of the signal based on variants selected by other PGS methods, for example lassosum or LDpred? I feel GenoBoost selects fewer SNVs overall given the constraint and some readers might incorrectly conclude from this that the more polygenic a trait is, the higher the proportion of SNVs with non-additive effects. This can be done in a few traits where the estimated h^2 is similar between GenoBoost and lassosum/LDpred. A plot of the number of SNVs included in each model would also be useful.

We agree about the importance of the careful interpretation of the modest correlation between polygenicity and the fraction of SNVs with inferred non-zero non-additive genetic dominance effects in GenoBoost (now reported in **Supplementary Fig 13a**). As you will see below, we now expanded our evaluation and considered the polygenicity and heritability across different metrics. We found that all of those measures support the modest correlation between the polygenicity and fraction of non-linear variant effects.

First, we investigated the number of SNVs with non-zero coefficients from other PGS models as a proxy measure of polygenicity. Specifically, we repeated the analysis focused on other PGS models and replicated a modest correlation of 0.65 in snpnet (**Supplementary Fig 13b**), 0.63 in snpboost (**Supplementary Fig 13c**), 0.59 in lassosum (**Supplementary Fig 13d**), and 0.28 in clumping and thresholding (**Supplementary Fig 13e**). We did not consider PGS models from LDpred, PRS-CS, and SBayesR, given that those PGS methods are designed to assign non-zero multivariate effect size estimates to all of the genetic variants, and the number of genetic variants included in the PGS models does not serve as a proxy measure of polygenicity.

Second, we also considered estimated liability-scale heritability without ascertainment correction (h^2) and correlated it with the fraction of SNVs with inferred non-zero non-additive genetic dominance effects. We found a modest correlation of 0.49 (**Supplementary Fig 13f**). Following the suggestion from the reviewer, we focused on 11 traits with similar estimated h^2 and repeated the analysis. We found a modest correlation of 0.35 after removing psoriasis, which has a larger estimated additive liability-scale heritability of 0.53 (**Supplementary Fig 13f**).

additive GenoBoost PGS model. The number of SNVs in the PGS models (x-axis) from Non-additive GenoBoost (a), snpnet (b), snpboost (c), lassosum (d), and C+T (e) and the proportion of SNVs with non-additive genetic dominance effects in the Non-additive GenoBoost PGS model (y-axis) are shown. The estimated additive heritability with standard error (x-axis) and the proportion of SNV with estimated non-additive genetic dominance effects (y-axis) is also shown (f). A gray line represents linear regression fit. We show Pearson's correlations in each comparison in the plot. For panel (f), we analyzed all phenotypes (gray) as well as eleven phenotypes psoriasis (orange). Source data are provided as a Source Data file.

Third, we now provide a supplementary data table containing the number of SNVs in PGS models and estimated heritability in **Supplementary Data 1**, as requested.

We subsequently updated the results section of the manuscript, addressing the important point raised by the reviewer.

Results (page 5, lines 245):

*We found a modest correlation (Pearson's correlation = 0.70) between the number of genetic variants selected in the GenoBoost PGS model and the fraction of the genetic variants classified as having non-additive genetic dominance effects (**Supplementary Fig 13a**). We also observed modest correlations when we used the number of genetic variants in other PGS models or the estimated additive liability-scale heritability (**Supplementary Fig 13b-f, Supplementary Data 1**).*

Overall, those additional analyses strengthen our confidence in the observed modest correlation between the polygenicity measure and the fraction of genetic variants with non-zero genetic dominance effects across the twelve traits analyzed in our study. Thank you very much for the suggestion.

3-i: I am a bit troubled by the results shown in Supplementary Figure 10, between a) and b) under the no dominance heritability scenarios. In a), when the overall h^2 is 0.1 and 100 causal variants, LDpred seems comparable to GenoBoost, however, in b) where the only difference is a higher h^2 ($=0.25$), then LDpred only captures 50% of the heritability ($\sim 8\%$) as compared to GenoBoost (h^2 estimate $\sim 16\%$). I wonder if there is some explanation for this. In line with this observation, I think the authors should be specific about that the simulation study did not incorporate LD, a scenario that puts LDpred at a disadvantage.

We appreciate the helpful comments and observations from the reviewer.

1. Difference in predictive performance between LDpred and GenoBoost in simulation study

We agree that the observed difference in the predictive performance between LDpred and GenoBoost in the original Supplementary Figure 10 is of concern. After careful investigation, we now acknowledge that one of the simulation parameter configurations used in the analysis was rather unrealistic and we have replaced the simulation analysis with the ones under more realistic parameter configurations.

Specifically, the parameter pair of ($h^2=0.25$, 100 causal genetic variants) imposes each simulated causal SNV to carry 2.5×10^{-3} ($=0.25/100$) of heritability. Indeed, when we performed

GWAS analysis on the synthetic phenotype, we observed that 96 SNVs have extreme p -values of $p < 1 \times 10^{-100}$, indicating the presence of strong local heritability in those regions. The substantial deviation of the genetic architecture of the simulated traits from the realistic phenotype is evident when we compare the GWAS association profile with the ones from disease outcomes in UK Biobank. For example, type 2 diabetes in UK Biobank (estimated $h^2=0.22$) shows more polygenic architecture. Furthermore, we found that LDpred shows convergence error under this unrealistic parameter configuration of ($h^2=0.25$, 100 causal genetic variants) and we believe it would explain the observed difference in the predictive performance.

In the revised manuscript, we performed additional simulation analysis under more realistic scenarios of $h^2=0.05$. We confirmed that the GWAS association analysis conducted for the revised parameter configurations, which have 0 to 6 SNVs with $p < 1e-100$, are more similar to that of disease outcomes in UK Biobank. As you will see below in **Supplementary Figure 10**, the difference in the predictive performance between LDpred and GenoBoost is much smaller in the revised analysis. Of note, we found that LDpred outperformed GenoBoost for $h^2=0.05$, $h_a^2=0.01$, and 1000 causal variants.

simulations per parameter are shown along with boxplots (n=10). On each box, the center line is the median, the top and bottom of the box are the first and third quartiles (Q3 and Q1), and the upper and lower whiskers are $Q3 + 1.5 \text{ IQR}$ and $Q1 - 1.5 \text{ IQR}$, respectively, where $\text{IQR} = Q3 - Q1$. Source data are provided as a Source Data file.

We clarified the observation in the revised manuscript.

Results (page 4, lines 177):

*To further test the effects of polygenicity on the predictive performance, we generated 80 synthetic phenotypes under eight simulation configurations with varying levels of polygenicity and heritability, applied GenoBoost and LDpred⁵, the best-performing method without the sparsity constraints, and compared their predictive performance (**Supplementary Fig 10, Supplementary Note 2**). GenoBoost outperformed LDpred for seven out of eight parameter configurations tested in our simulation analysis. The difference in the predictive performance between GenoBoost and LDpred is larger in less polygenic scenarios.*

2. LD structure in the simulation study

In our simulation analysis, we used the genotype data in UK Biobank and prepared synthetic phenotypic data. As such, the LD structure is present in our simulation settings. We acknowledge that none of the simulation studies, including ours, is perfect. We believe further simulation study, perhaps with more recently developed genotype simulators (for example, HAPNEST described in Wharrie et al. 2023. PubMed ID: 37647640), is of interest for future studies.

We believe that those additional analyses have strengthened our simulation study and helped better illustrate the advantages of GenoBoost under the presence of genetic dominance effects.

Minor comments:

- Figure labels and legends incorrect for Supplementary Figure 10: $h^2=0.1$ in a) and $h^2=0.25$ in b)
- Legend of Supplementary Figure 12 '(n=2,975; African, n=2,867; South Asian, n=904)': it is not clear which sample size for which population. I suspect East Asian is for the last one?
- Given these minor inconsistencies, I suggest the authors have a final check on all figure legends and tables.

We thank the reviewer for catching and pointing out this mistake. We now have fixed these errors and confirmed all the legends match with the corresponding main and supplementary display items.

We thank the reviewer for the kind words and the appreciation of our revised manuscript.

Reviewer #3 (Remarks to the Author):

The authors did a great job mapping different aspects of the non-additive models and their strengths and limitations. The positive correlation between polygenicity and # of non-additive effects is unexpected yet interesting and are specific to traits studied here. I have no further comments.

Reviewer #3 (Remarks on code availability):

They run well and the program was easy to set up.